# Sex differences in heart mitochondria regulate diastolic dysfunction

Yang Cao[1], Laurent Vergnes[2,3], Yu-Chen Wang[1], Calvin Pan [1], Karthickeyan Chella Krishnan[1,4], Timothy M. Moore [1], Manuel Rosa-Garrido [5], Todd H. Kimball[6], Zhiqiang Zhou[1], Sarada Charugundla[1], Christoph D. Rau[6], Marcus M. Seldin [1], Jessica Wang [1], Yibin Wang [6,7,8], Thomas M. Vondriska[6,7,8], Karen Reue[2,3,9] & Aldons J. Lusis [1,3,10 ✉]

Heart failure with preserved ejection fraction (HFpEF) exhibits a sex bias, being more common in women than men, and we hypothesize that mitochondrial sex differences might underlie this bias. As part of genetic studies of heart failure in mice, we observe that heart mitochondrial DNA levels and function tend to be reduced in females as compared to males. We also observe that expression of genes encoding mitochondrial proteins are higher in males than females in human cohorts. We test our hypothesis in a panel of genetically diverse inbred strains of mice, termed the Hybrid Mouse Diversity Panel (HMDP). Indeed, we find that mitochondrial gene expression is highly correlated with diastolic function, a key trait in HFpEF. Consistent with this, studies of a "two-hit" mouse model of HFpEF confirm that mitochondrial function differs between sexes and is strongly associated with a number of HFpEF traits. By integrating data from human heart failure and the mouse HMDP cohort, we identify the mitochondrial gene *Acsl6* as a genetic determinant of diastolic function. We validate its role in HFpEF using adenoviral over-expression in the heart. We conclude that sex differences in mitochondrial function underlie, in part, the sex bias in diastolic function.

[1] Division of Cardiology, Department of Medicine, University of California, Los Angeles, CA, USA. [2] Metabolism Theme, David Geffen School of Medicine at UCLA, Los Angeles, CA 90095, USA. [3] Department of Human Genetics, David Geffen School of Medicine at UCLA, Los Angeles, CA 90024, USA. [4] Department of Pharmacology and Physiology, University of Cincinnati College of Medicine, Cincinnati, USA. [5] Department of Biomedical Engineering, School of Medicine and School of Engineering, University of Alabama at Birmingham, Birmingham, AL 35233, USA. [6] Department of Anesthesiology & Perioperative Medicine, David Geffen School of Medicine at UCLA, Los Angeles, CA, USA. [7] Department of Medicine, David Geffen School of Medicine at UCLA, Los Angeles, CA, USA. [8] Department of Physiology, David Geffen School of Medicine at UCLA, Los Angeles, CA, USA. [9] Molecular Biology Institute at UCLA, Los Angeles, CA 90095, USA. [10] Department of Microbiology, Immunology and Molecular Genetics, University of California, Los Angeles, CA, USA. ✉email: jlusis@mednet.ucla.edu

HFpEF is an increasingly prevalent syndrome that is characterized by diastolic dysfunction and preserved ejection fraction. It is distinct from heart failure with reduced ejection fraction (HFrEF) in terms of pathogenesis and effective therapeutic management[1–3]. Epidemiologic data indicate that HFpEF accounts for half of all cases of HF with multiple comorbidities including diabetes, hypertension, and restrictive cardiomyopathies[4]. Individuals with systolic heart failure also present with features of diastolic dysfunction, emphasizing diastolic dysfunction as an important contributor to the morbidity of heart failure[5]. Drugs that are effective in HFrEF have no beneficial effects on morbidity and mortality in HFpEF[1,6–8]. The molecular mechanisms underlying HFpEF are poorly understood, and we have used genetic approaches in mouse models to help identify genes and pathways contributing to the disorder.

One of the questions we have addressed is the role of sex differences in HFpEF. Consistent epidemiological data demonstrate that women are about twice as likely to develop HFpEF as compared to men and that they tend to present with more symptom burden, including significantly increased diastolic dysfunction and enhanced left ventricle stiffness[9–12]. However, HFpEF is highly heterogeneous and the notion of sex differences in HFpEF has been called into question[1,13–15]. Compared to men, women have structural and functional differences in the heart including smaller left ventricle (LV) chambers, lower stroke volumes, more prominent concentric remodeling upon pressure overload, higher systolic and diastolic LV stiffness at a given age, and more profound response to hypertension and obesity[16,17]. One approach to help resolve the issue is to examine mouse models where genetic and environmental factors can be controlled.

Our genetic studies of heart failure traits in a diverse cohort of mice revealed a sex bias in the content of mitochondrial DNA and in mitochondrial function, with males exhibiting higher levels of cardiomyocytes. Given the extensive evidence for the role of mitochondria in heart failure, we hypothesized that the sex differences in mitochondria may contribute in part to greater susceptibility in women. We have examined this hypothesis using several approaches, and using genetics we identified a mitochondrial protein, ACSL6, that appears to protect against various HFpEF traits, including diastolic dysfunction, in a "two-hit" mouse model of the disorder.

## Results

**Study design**. We sought to investigate genetic factors, including sex differences, that contribute to traits such as diastolic and metabolic dysfunction in HFpEF. This is difficult to address directly in human populations due to genetic heterogeneity, inability to access relevant tissues, and complex environmental factors. For our studies, we selected the HMDP, a cohort of more than 100 genetically diverse inbred mouse strains that allows high-resolution genetic mapping, replication, and multi-omics analyses in the context of genetics[18–20]. We also sought to examine the relevance of our findings to human disease by interrogating publicly available human data.

**Sex differences in diastolic dysfunction in HFpEF**. To test whether females are more susceptible to diastolic dysfunction in the progression of HFpEF, we reproduced the "two-hit" HFpEF mouse model in C57BL/6 J male and female mice (Fig. 1A). The model recapitulates clinical phenotypes of HFpEF after 7 weeks of high-fat diet (HFD) + l-NAME feeding. Both male and female mice exhibited a significant increase in body weight, fat mass, left ventricle mass, and adipose weight, as well as a decrease in exercise tolerance (Supplementary Fig. 1A-1F). HFD + l-NAME

feeding also induced glucose intolerance in mice of both sexes (Supplementary Fig. 1G-1H). In addition, fatty-acid-binding protein 3 (*Fabp3*), an early biomarker of cardiac damage, was increased in HFpEF hearts relative to chow diet controls, indicating the dysregulation in the heart (Fig. 1B). Compared with male mice, female mice exhibited higher E/A ratios, E/e' ratios, and lung weights (Fig. 1C-E). In contrast, left ventricle ejection fraction (LVEF) was preserved under HFD + l-NAME diet (Fig. 1F).

We note that investigators in a previous study of the "two-hit" model, using the inbred mouse substrain C57BL/6 N, observed that males exhibited increased susceptibility to HFpEF traits[21]. Our HFpEF model studies were performed on a C57BL/6 J genetic background, and we assume that the difference in sex bias is due to genetic background, although experimental differences could be involved. We previously showed that there are clear gene-by-sex interactions among the HMDP, with the effect of sex being dependent on the genetic background[22]. We have begun to examine the HFpEF "two-hit" model in the entire HMDP, and results with about 30 strains thus far show a significant increase in susceptibility in females (Supplementary Fig. 1I).

**Diastolic function is associated with mitochondrial DNA levels in an isoproterenol model of heart failure**. To identify genetic factors involved in diastolic function, we analyzed data from a previously published study of isoproterenol-induced cardiomyopathy in 105 inbred strains of female HMDP mice (ISO-HMDP), focusing on the trait of diastolic function. In that study, adult female mice were treated with ISO infusion for 21 days and cardiac function was determined weekly by echocardiography (Fig. 1G). After 21 days of ISO treatment, global transcriptome profiling of left ventricles was performed. We used E/A ratio (the ratio of peak velocity blood flow in early diastole to peak velocity flow in late diastole), heart weight, and lung weight as surrogates for cardiac diastolic function. We performed gene-by-trait correlation analysis, identifying 1313 and 1875 genes that were significantly correlated with E/A ratio at baseline and 21 days after ISO treatment (filtered by adjusted $p$-value < 0.01), respectively. Genes that were significantly correlated with E/A ratio were enriched in pathways related to mitochondrion, cell–cell adherens junction, protein transport, TCA cycle, autophagy, and mitochondrion inner membrane (Fig. 1H). These data suggest that alterations in mitochondrial metabolism underly changes in diastolic function, consistent with previous reports that cardiomyocyte mitochondria in HFpEF have structural and energetic abnormalities[23,24].

To directly examine the relationship of mitochondrial levels to cardiovascular traits, we quantitated mitochondrial copy number in the ventricles of the female ISO-HMDP mice and performed trait-by-trait correlations. Notably, mtDNA content was inversely correlated with multiple cardiac traits, including lung mass/body weight, left atrium mass/body weight, IVS at end-diastole (day 0), and IVS to PW ratio at end-diastole (day 0) (Fig. 1I-L), suggesting that mitochondria contribute to diastolic properties. We also reanalyzed the data following the removal of samples with extremely high mtDNA content. With the exception of mitral inflow E velocity, the correlations remained significant (Supplementary Fig. 2).

**Mitochondrial DNA levels and gene expression exhibit a sex bias, with males tending to show a higher expression than females in both mice and humans**. We measured heart mitochondrial DNA (mtDNA) copy number in 100 strains of male and female mice of the HMDP maintained on a Western (high fat and sucrose) diet. We used PCR to determine the copy number of

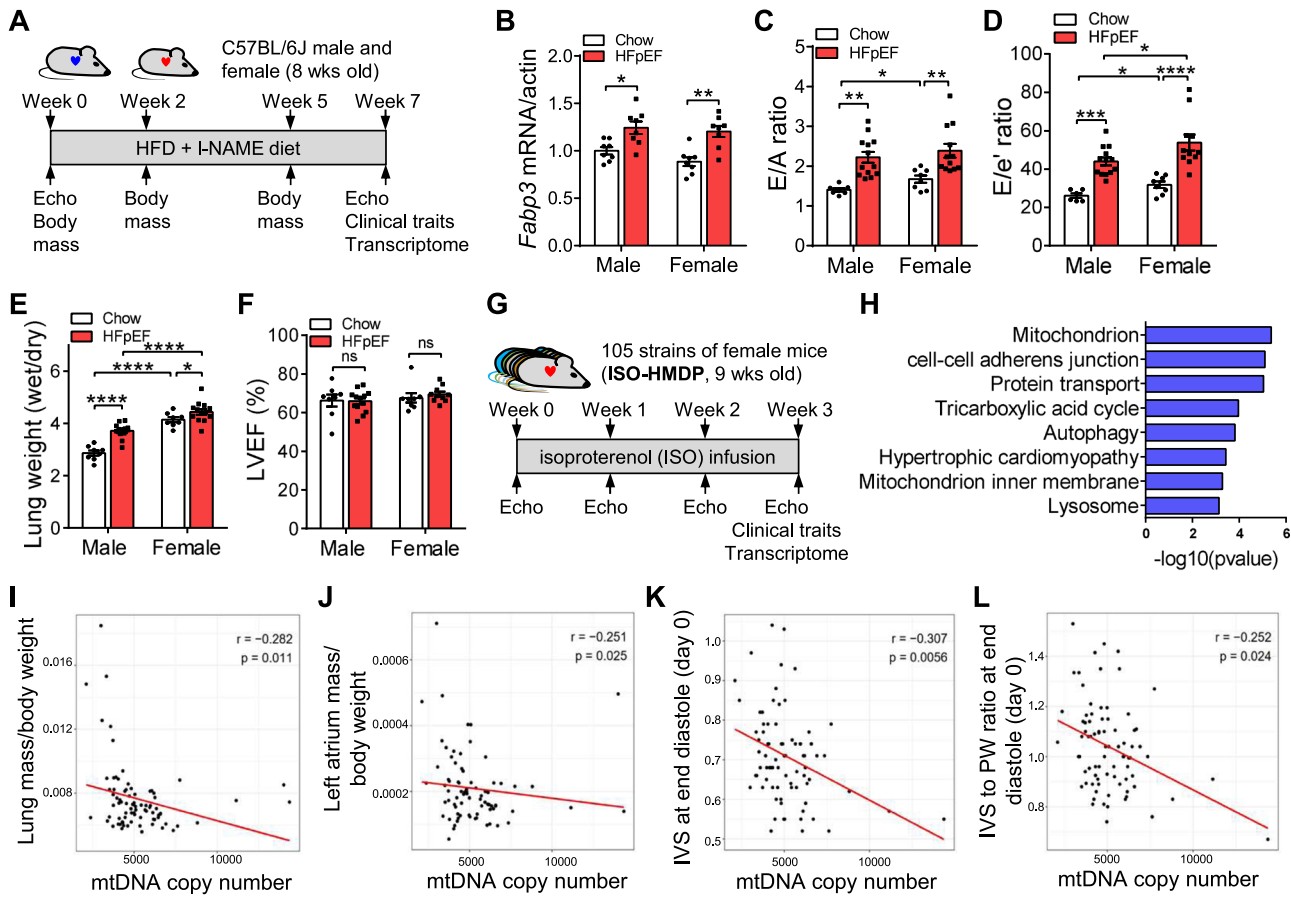

**Fig. 1 Sex-biased diastolic dysfunction in the progress to HFpEF and mtDNA level was associated with diastolic function. A** Experimental timeline for the development of HFpEF in C57BL/6 J mice (8 weeks old). **B** *Fabp3* mRNA level in C57BL/6 J male and female heart after 7 weeks of HFD + I-NAME diet. N = 8. **C–F** E/A ratio (**C**), E/e' ratio (**D**), wet/dry ratio of lung weight (**E**), and left ventricle ejection fraction (**F**) of C57BL/6 J male and female mice after 7 weeks of HFD + l-NAME diet. Male-chow, n = 7 (**C**), 6 (**D**), or 8 (**E**, **F**); male-HFpEF, n = 13 (**C**, **D**, **F**) or 12 (**E**); female-chow, n = 8; female-HFpEF, n = 12 (**C**), 11 (**D**), 13 (**E**), or 10 (**F**). **B**, p (male) = 0.016; p (female) = 0.001. **C** p (male) = 0.002; p (female) = 0.007; p (male–female) = 0.03. **D** p (male) = 0.001; p (female) < 0.0001; p (male–female) = 0.03 (chow) and 0.04 (HFpEF). **E** p (male) < 0.0001; p (female) = 0.04; p (male–female) < 0.0001 (chow) and < 0.0001 (HFpEF). **G**. Experimental design for the development of isoproterenol (ISO) induced cardiomyopathy across HMDP mice. Nine-week-old female mice from 105 of the HMDP inbred strains were treated with isoproterenol (30 mg/kg/day) via an intra-abdominally implanted osmotic pump for 21 days. Heart function at baseline (prior to ISO pump implantation), week 1, week 2, and week 3 were obtained with echocardiography. After 3 weeks of ISO infusion, mice were sacrificed and left ventricle transcriptome was performed. N = 3–12 for each strain. **H** Enriched pathways of genes that were significantly associated with mitral inflow E to A velocity ratio (E/A ratio) across female ISO-HMDP cohort. Pathway analysis was performed with The Database for Annotation, Visualization, and Integrated Discovery (DAVID)[49]. Scores from DAVID are modified Fisher tests, corrected for false discovery rate (FDR) using the Benjamini–Hochberg (BH) method. **I–L** mtDNA copy number significantly correlated with lung mass/body weight (**I**), left atrium mass/body weight (**J**), IVS at end-diastole (day 0, **K**), and IVS to PW ratio at end-diastole (day 0, **L**) in ISO-HMDP mice. Each point represents a mouse from an inbred strain. IVS interventricular septal thickness, PW posterior wall thickness. p-values are from biweight midcorrelation (bicor) tests. Each point represents a mouse. All data are presented as the mean ± SEM. ns not significant, *P < 0.05, **P < 0.01, ***P < 0.001, and ****p < 0.0001 by two-way ANOVA (**B–F**). Source data are provided as a Source Data file.

mitochondria per cell[25,26]. We observed that, across the HMDP, the hearts of females exhibited significantly reduced mtDNA content (copies per cell) compared with males (Fig. 2A)[27]. We also examined the expression of mitochondrial genes that are part of the mitochondrial genome or nuclear genes that encode mitochondrial proteins in the heart. As shown in volcano plots, males tended to have higher levels of mitochondrial gene expression (Fig. 2B), including expression of electron transport chain (ETC) genes (Fig. 2C). After normalization to mtDNA content in each strain, mitochondrial genes and ETC genes remained higher in males (Supplementary Fig. 3).

Cordero et al. previously reported human cardiac gene expression networks from 177 failing hearts (145 male and 32 female hearts) and 136 healthy hearts (73 male and 63 female hearts)[28]. To profile mitochondrial gene expression in male and

female human hearts, we examined the expression of mitochondrial genes and ETC genes in these samples. A clear bias was observed, with male hearts exhibiting higher levels of most mitochondrial genes relative to females in all samples (failing and healthy) and healthy samples alone (Fig. 2D-E). Consistent with these results, ETC genes in both healthy and failing hearts were reduced in female hearts vs. male hearts (Fig. 2F-G).

**Heart mitochondrial DNA and functions are regulated by sex hormones**. To determine whether the sex differences are due to hormonal or chromosomal effects, we performed gonadectomy in male and female mice of the C57BL/6 J strain. Notably, gonadectomy reduced heart mtDNA content in males, whereas it increased mtDNA in females (Fig. 3A-B). Hormone replacement indicated that testosterone increased mitochondrial copy number

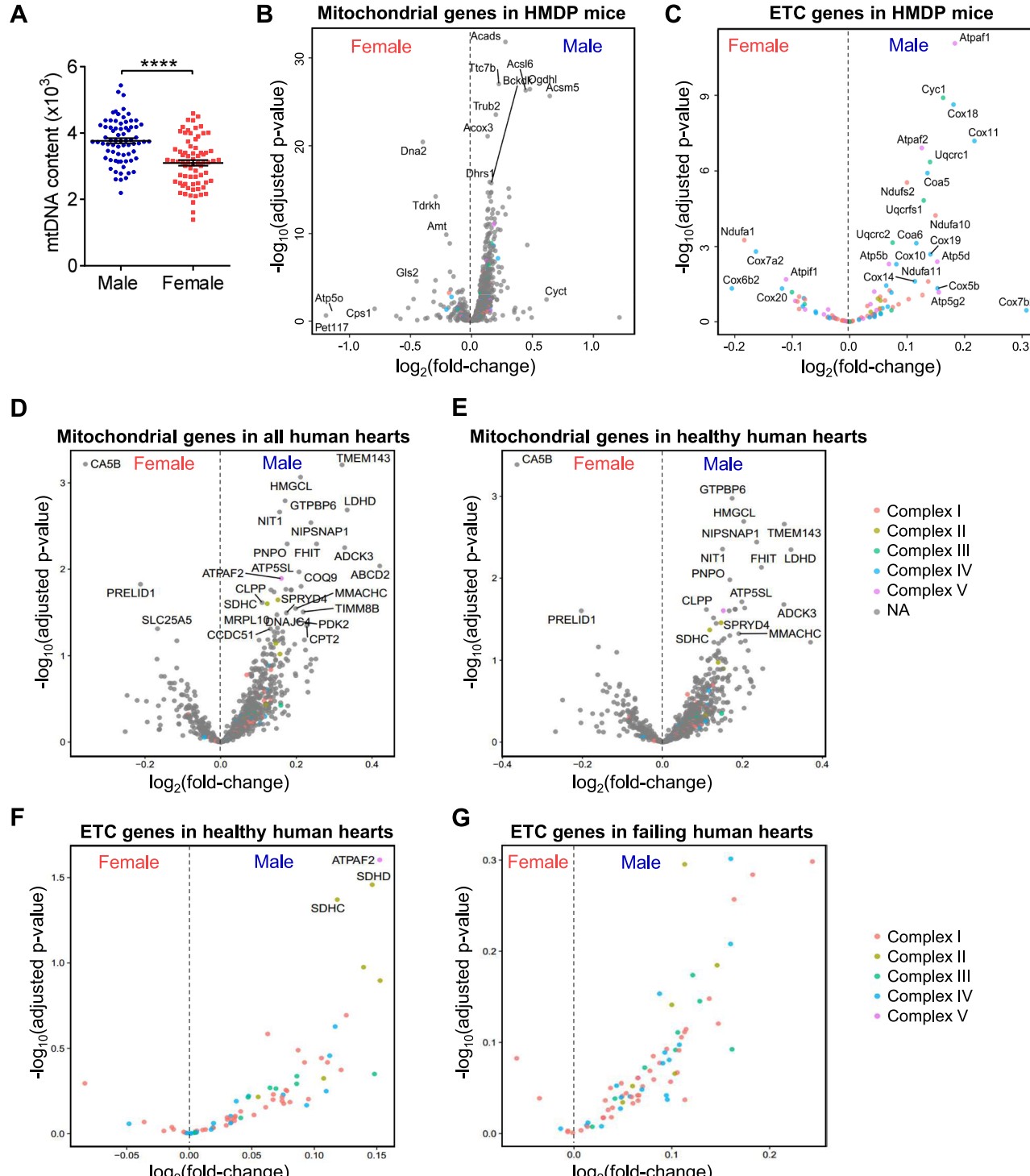

**Fig. 2 Mitochondrial gene expression in human and mouse hearts. A** Heart mtDNA copy number in male and female HMDP mice (about 10 weeks old). Each point represents a mouse from an inbred strain. $N = 72$. $P < 0.0001$. **B**, **C** Differential expressed genes encoding mitochondrial proteins (**B**) and electron transport chain (ETC) proteins (**C**) in 100 strains of HMDP mice. **D**, **E** Differential expressed mitochondrial genes in all human hearts (**D**) and healthy human hearts (**E**). **F**, **G** Differential expressed ETC genes in healthy (**F**) and failing (**G**) human hearts. Genes of each complex were denoted. All data are presented as the mean ± SEM. ****$P < 0.0001$ by two-sided Student's $t$-test (**A**). DESeq2 results use the Wald test, corrected for false discovery rate (FDR) using the Benjamini–Hochberg (BH) method (**B**–**G**). Source data are provided as a Source Data file.

in males and that estrogen reduced mitochondrial copy number in females. To further determine heart mitochondrial function, we reproduced a "two-hit" HFpEF mouse model[29] in C57BL/6 J male and female mice after gonadectomy (Fig. 3C and Supplementary Fig. 4). Gonadectomy in male mice reduced body weight

and lean mass (Supplementary Fig. 4A-4C), whereas it increased body weight, fat mass, and lean mass in female mice under HFD + l-NAME feeding[30] (Supplementary Fig. 4D-4F). Consistent with the change of mtDNA content, gonadectomy reduced the activity of state 3 u (maximal respiration), complex I, II, IV,

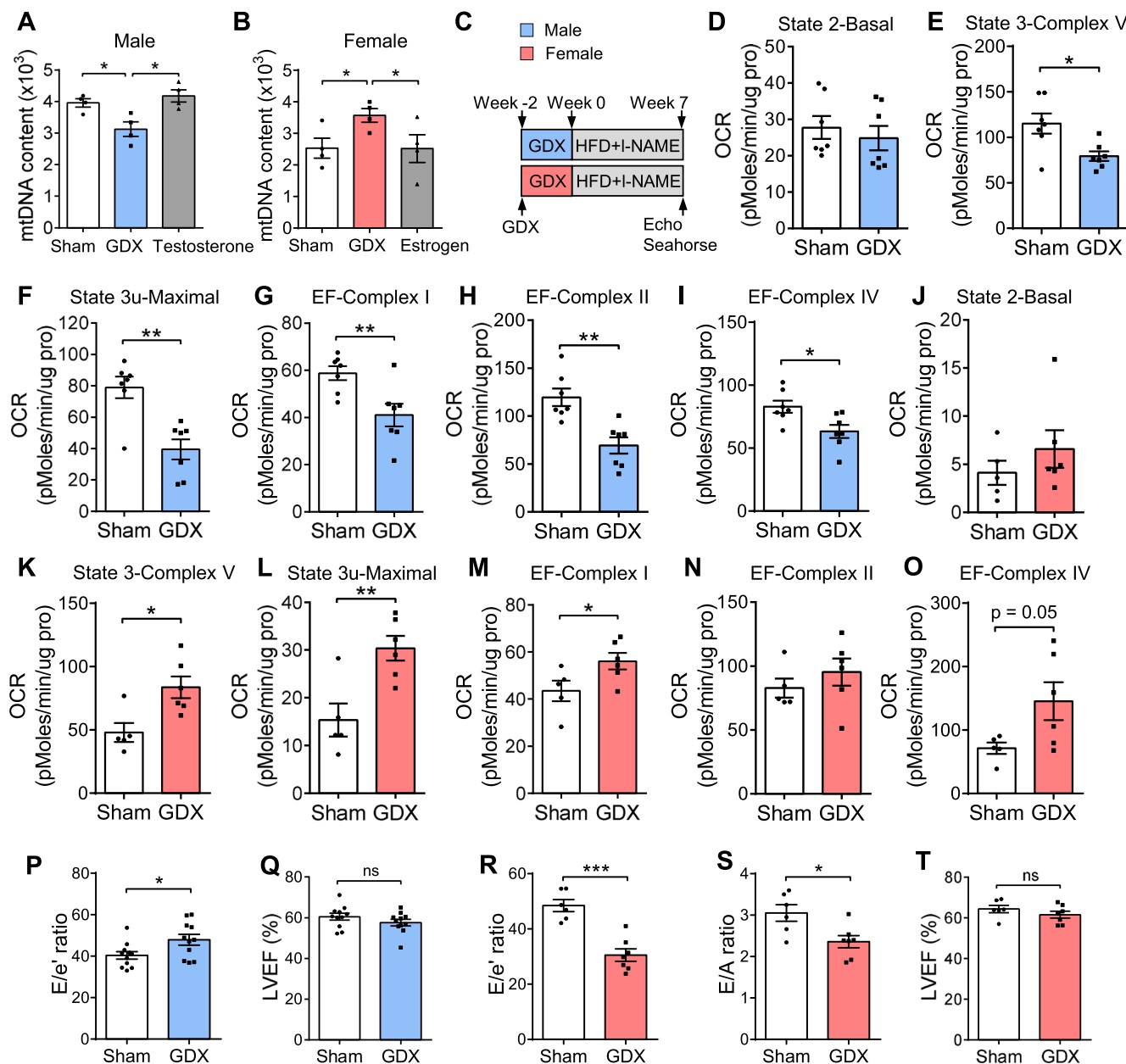

**Fig. 3 Mitochondrial and diastolic function in gonadectomized mice. A, B** Heart mtDNA copy number in C57BL/6 J male (**A**) and female (**B**) mice subjected to sham, gonadectomy or hormone replacement after gonadectomy under chow diet. mtDNAs were normalized to nuclear DNA. $N = 4$. **A** $p = 0.019$ (Sham/GDX) and 0.013 (GDX/Testosterone). **B** $p = 0.036$ (Sham/GDX), and 0.04 (GDX/Estrogen). **C** Experimental timeline for the gonadectomy and induction of HFpEF in C57BL/6 J male and female mice. **D–I** Oxygen consumption rates (OCR) of isolated mitochondria from C57BL/6 J male hearts were measured by Seahorse assay. The coupling assay measures basal respiration in presence of palmitoyl-carnitine, state 3 (ADP) and state 3 u (FCCP). The electron flow measures complex I (palmitoyl-carnitine), complex II (succinate), and complex IV (TMPD) respiration. $N = 7$. **E** $p = 0.012$. **F** $p = 0.001$. **G** $p = 0.008$. **H** $p = 0.001$. **I** $p = 0.017$. **J–O** Same Seahorse assays were performed in C57BL/6 J female mice after gonadectomy and 7 weeks of HFD + l-NAME feeding. Sham, $n = 5$; GDX, $n = 6$. **K** $p = 0.013$. **L** $p = 0.006$. **M** $p = 0.049$. **P, Q** E/e' ratio (**P**) and LVEF (**Q**) of C57BL/6 J male mice after gonadectomy and 7 weeks of HFD + l-NAME diet. $N = 11$. **P** $p = 0.027$. **R–T** E/e' ratio (**R**), E/A ratio (**S**), and LVEF (**T**) of C57BL/6 J female mice after gonadectomy and 7 weeks of HFD + l-NAME diet. Sham, $n = 6$; GDX, $n = 7$. **R** $p = 0.0001$. **S** $p = 0.016$. Each point represents a mouse. All data are presented as the mean ± SEM. ns not significant, *$P < 0.05$, **$P < 0.01$, and ***$P < 0.001$, by one-way ANOVA (**A**, **B**) or by two-sided Student's t-test (**D–T**). Source data are provided as a Source Data file.

and complex V in males (Fig. 3D-I), whereas it increased these activities in females (Fig. 3J-O).

**Hormonal regulation of diastolic function in HFpEF.** As shown above, sex hormones regulate mitochondrial functions that are, in turn, associated with diastolic function. We therefore hypothesized that sex hormones may have an impact on diastolic performance. To confirm this, we performed gonadectomy in C57BL/6 J male and female mice, followed by HFD + l-NAME feeding for 7 weeks (Fig. 3C). Compared with the sham operation, gonadectomy caused an increase in E/e' ratio while LVEF was preserved in male mice upon HFpEF development (Fig. 3P-Q). In contrast, gonadectomy reduced E/e' ratio and E/A ratio in female mice (Fig. 3R-T), demonstrating that sex hormones affect diastolic performance in the HFpEF model.

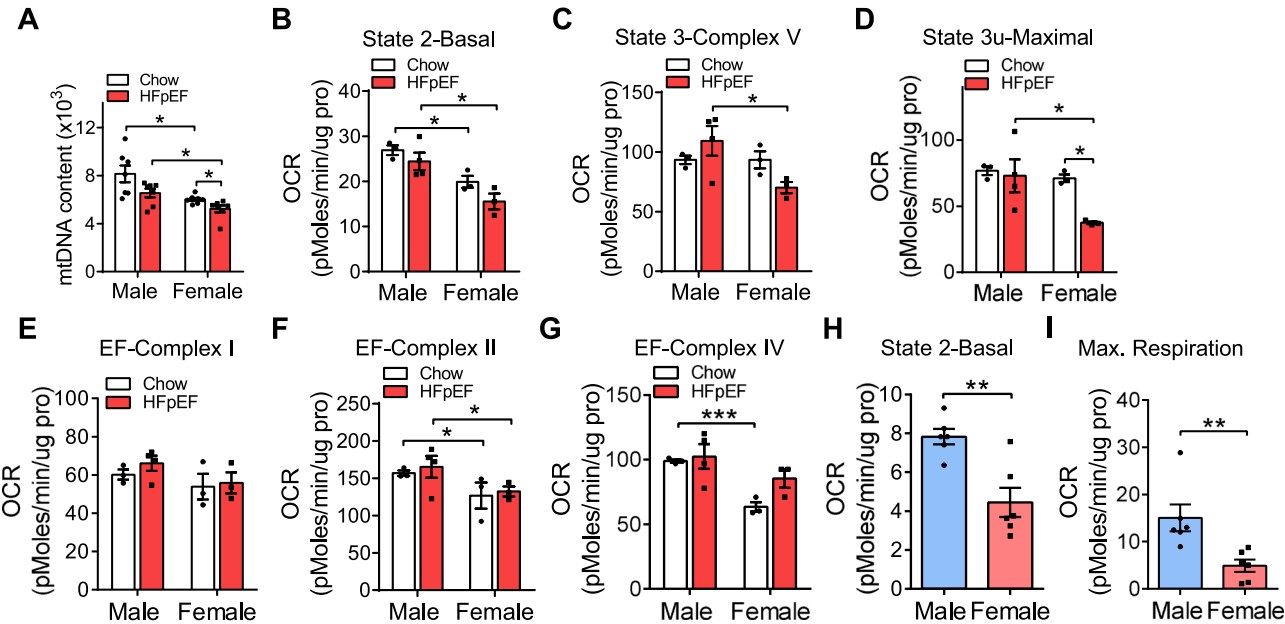

**Fig. 4 Female mice exhibited reduced mitochondrial function. A** Heart mtDNA content in C57BL/6 J male and female mice (8 weeks old) fed with chow diet or HFD + l-NAME for 7 weeks. N = 7. p (male-chow/female-chow) = 0.011; p (male-HFpEF/female-HFpEF) = 0.015; p (female) = 0.03. **B–G** Oxygen consumption rates (OCR) of isolated mitochondria from C57BL/6 J male and female hearts were measured by Seahorse assay. The coupling assay measures basal respiration in presence of palmitoyl-carnitine, state 3 (ADP), and state 3 u (FCCP). The electron flow measures complex I (palmitoyl-carnitine), complex II (succinate), and complex IV (TMPD) respiration. Chow, n = 3; HFpEF, n = 4 (male) or 3 (female). **B** p (male-chow/female-chow) = 0.014; p (male-HFpEF/female-HFpEF) = 0.022. **C** p (male-HFpEF/female-HFpEF) = 0.046. **D** p (male-HFpEF/female-HFpEF) = 0.048; p (female-chow/female-HFpEF) = 0.01. **F** p (male-chow/female-chow) = 0.026; p (male-HFpEF/female-HFpEF) = 0.043. **G** p (male-chow/female-chow) = 0.0009. **H–I** Adult cardiomyocytes were isolated from male and female mice after 7 weeks of HFD + l-NAME feeding. OCR was measured before and after the sequential injection of 1.5 μM oligomycin, 1.5 μM FCCP, and 4 μM of rotenone/myxothiazol. N = 6. **H** p = 0.002. **I** p = 0.009. Each point represents a mouse. All data are presented as the mean ± SEM. *P < 0.05, **P < 0.01, and ***P < 0.001, by two-way ANOVA (**A–G**) or by two-sided Student's t-test (**H–I**). Source data are provided as a Source Data file.

**Mitochondrial function in male and female mice in the HFpEF "two-hit" model.** Next, we examined mtDNA content in C57BL/6 J male and female mice fed with a chow diet or HFD + l-NAME diet for 7 weeks. Compared with male hearts, mtDNA was lower in female hearts, and it was further decreased after HFpEF induction (Fig. 4A). To directly examine mitochondrial function in both sexes in the context of HFpEF, we isolated mitochondria from the hearts of male and female mice under chow diet and under HFpEF conditions and provided palmitoyl-carnitine as a substrate for mitochondrial respirometry assays. Basal, state 3 (in presence of ADP), and state 3 u (maximal) respiration in the coupling assay were substantially reduced in mitochondria from female compared to male hearts (Fig. 4B–D), demonstrating that the function of mitochondria in females was impaired during the progression of HFpEF. In addition, mitochondria in females exhibited lower activity of complex II and complex IV relative to the male group in the electron flow assay (Fig. 4E–G). Since mitochondria are highly enriched in cardiomyocytes in the heart, we suspected that the sex differences in mitochondrial function were due to cardiomyocytes. To test this, we isolated adult cardiomyocytes from male and female mice and examined mitochondrial function. As expected, cardiomyocytes isolated from female hearts exhibited reduced basal and maximum respiration relative to males (Fig. 4H-I), demonstrating reduced mitochondrial function in female hearts.

**Identification of _Acsl6_ as a sex-biased _cis_-regulator of diastolic function.** To investigate the underlying mechanism of impaired diastolic performance in females, we sought to identify the molecular signatures differing between males and females. We performed RNA-seq for heart tissue collected from C57BL/6 J male and female mice fed with a chow diet and HFD + l-NAME diet. The principal component analysis (PCA) of differentially expressed genes (DEGs) exhibited four main clusters between male and female mice under chow and HFpEF conditions (Fig. 5A), suggesting sex- and disease-biased changes in transcriptome profiles. Our analysis identified 2770 DEGs discriminating between HFpEF and Chow conditions in males (Supplementary Fig. 5A) and 1714 DEGs in females (Supplementary Fig. 5B). Among the 1714 DEGs, 775 genes were also changed in males while the rest were specific for females, defining a sex-biased molecular signature for HFpEF. Sex-biased genes were enriched in pathways related to inflammation, cholesterol efflux, serine synthesis, cell cycle, insulin signaling, fatty-acid metabolism, and heart development (Supplementary Fig. 5C).

We also sought to identify sex-biased regulators of diastolic function by overlapping the gene expression data in female ISO-HMDP (Fig. 2B-C) and "two-hit" HFpEF cohorts (Supplementary Fig. 5), as well as the human heart failure cohort (Fig. 2D-G)[28]. In the human heart failure cohort, sex-biased DEGs were enriched in pathways including cell differentiation and mitochondrion (Supplementary Fig. 6A). Among the sex-biased genes, _Acsl6_ (Acyl-CoA Synthetase Long Chain Family Member 6) expression was significantly lower in female hearts compared with males across the HMDP (Fig. 5B) and was further decreased after HFpEF development (Fig. 5C and D). Gonadectomy reduced _Acsl6_ expression in male hearts whereas it increased _Acsl6_ expression in female hearts (Fig. 5E), consistent with a contribution to increased mitochondrial and metabolic dysfunction in HFpEF mice. Similarly, _Fasn_ and _Acaca_ were decreased in female hearts relative to male hearts (Supplementary Fig. 6B-6C), supporting the possibility that impaired fatty-acid metabolism

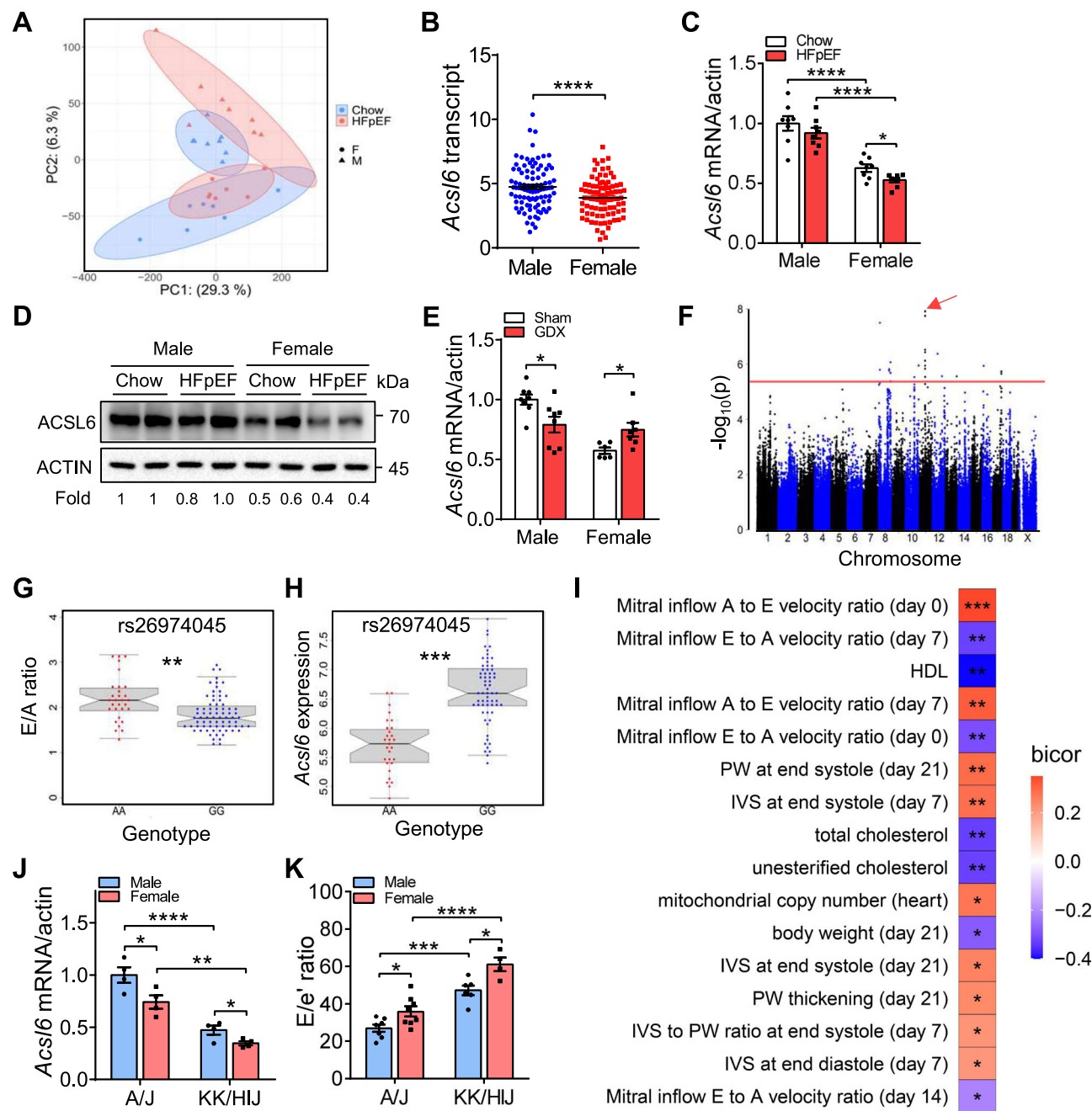

contributes to increased susceptibility to HFpEF in female mice. While not located near the genome-wide association study (GWAS) of E/A in female ISO-HMDP (Supplementary Fig. 7), expression quantitative trait locus (eQTL) analysis of *Acsl6* in female ISO-HMDP revealed that it is regulated locally (*cis*-eQTL, *p*-value = 1.22E-08) (Fig. 5F and Supplementary Fig. 8). Further, we observed a lower E/A ratio and higher *Acsl6* expression with the GG allele of the peak locus SNP (rs26974045, Fig. 5G-H). Consistent with a causal role, local genetic variation in *Acsl6* expression was significantly correlated with key traits of diastolic function including E/A ratio, mtDNA copy number, and body weight (Fig. 5I and Supplementary Fig. 9), suggesting that *Acsl6* may be causal for diastolic function[31].

*Acsl6* expression exhibits wide variation across strains in the female ISO-HMDP (Supplementary Fig. 10A). We examined diastolic function in two strains with dramatic differences in *Acsl6* expression: A/J and KK/HIJ (Fig. 5J and Supplementary Fig. 10A).

We developed the "two-hit" HFpEF model in male and female mice of these two strains. The KK/HIJ strain showed dramatically higher levels of E/e' ratio (Fig. 5K) and glucose intolerance (Supplementary Fig. 10B) in both male and female mice compared with A/J, consistent with our hypothesis.

**Acsl6 protects against diastolic and metabolic dysfunction in a sex-biased manner.** To validate the impact of the *Acsl6* candidate gene on diastolic function, we overexpressed *Acsl6* in the hearts of C57BL/6 J male and female mice using an adeno-associated virus serotype 9 vector carrying *Acsl6* under the regulation of the cardiac-specific cTnT promoter (AAV9-*Acsl6*) (Fig. 6A). We induced HFpEF in AAV9-GFP and AAV9-*Acsl6* male and female mice to test whether a higher *Acsl6* level alleviates diastolic dysfunction (Fig. 6A). In male mice, *Acsl6* expression was specifically increased in the heart relative to other tissues, including the liver, adipose, skeletal muscle, and kidney (Fig. 6B). Body weight and

**Fig. 5 Identification of *Acsl6* as a sex-biased *cis*-regulator of diastolic dysfunction. A** Principal component analysis (PCA) plot of male and female C57BL/6 J mice under chow diet and 7 weeks of HFD + l-NAME diet. DEGs were identified with a false discovery rate (FDR) < 0.05. **B** *Acsl6* mRNA levels in the hearts of male ($n = 96$) and female ($n = 87$) HMDP mice. $p < 0.0001$. **C, D** qRT-PCR (**C**) and western blotting (**D**) showing *Acsl6* mRNA and protein in C57BL/6 J male and female hearts under chow and HFpEF condition. mRNA and protein levels were normalized to actin. $N = 8$. **C** $p$ (male-chow/female-chow) < 0.0001; $p$ (male-HFpEF/female-HFpEF) < 0.0001; $p$ (female-chow/female-HFpEF) = 0.014. **E** qRT-PCR showing mRNA levels of *Acsl6* in C57BL/6 J male and female heart after gonadectomy and 7 weeks of HFD + l-NAME diet. mRNA levels were normalized to actin. Male, $n = 8$; female, $n = 6$ (sham) or 7 (GDX). $p$ (male) = 0.019; $p$ (female) = 0.024. **F** Manhattan plot showing the significance ($-\log10$ of p) of all SNPs and eQTL of *Acsl6* in female mice after 3 weeks of isoproterenol infusion (female ISO-HMDP). The significant threshold (red line) of $p = 4.1 \times 10^{-6}$ is indicated. Association p-values from FaST-LMM are from Wald tests uncorrected for FDR. **G, H** E/A ratio distribution (**G**) and *Acsl6* heart expression (**H**) based on genotype distribution at peak SNP associated with *Acsl6* on chromosome 11 (rs26974045). Box and whisker plot depicting mean and distribution. Boxplots are shown centered on medians with upper and lower quartiles indicated by the box boundaries; whiskers enclose the box ± 1.5× the interquartile range (IQR). AA allele, $n = 30$; GG allele, $n = 75$. **G** $p = 0.005$. **H** $p = 0.0002$. **I** The correlation between local genetic variation in *Acsl6* expression (cis-eQTL) and parameters of diastolic function and metabolism in ISO-HMDP. $p$-values are from biweight midcorrelation (bicor) tests. **J, K** *Acsl6* expression (**J**) and E/e' ratio (**K**) in male and female A/J and KK/HIJ strains of mice after 7 weeks of HFD + l-NAME feeding. **J** $n = 4$. **K** A/J-male, $n = 7$; A/J-female, $n = 8$; KK/HIJ-male, $n = 6$; KK/HIJ-female, $n = 4$. **J** $p$ (A/J-male/A/J-female) = 0.025; $p$ (A/J-male/KK/HIJ-male) < 0.0001; $p$ (A/J-female/KK/HIJ-female) = 0.001; $p$ (KK/HIJ-male/KK/HIJ-female) = 0.038. **K**, $p$ (A/J-male/A/J-female) = 0.025; $p$ (A/J-male/KK/HIJ-male) = 0.0001; $p$ (A/J-female/KK/HIJ-female) < 0.0001; $p$ (KK/HIJ-male/KK/HIJ-female) = 0.021. Each point represents a mouse. All data are presented as the mean ± SEM. $*P < 0.05$, $**P < 0.01$, $***P < 0.001$, and $****p < 0.0001$, by two-way ANOVA (**C, E, J, K**) or by two-sided Student's $t$-test (**B, G, H**). Source data are provided as a Source Data file.

fat mass were reduced by *Acsl6* overexpression after 7 weeks of HFD + l-NAME feeding (Fig. 6C). Food intake was not significantly affected by *Acsl6* overexpression (Supplementary Fig. 11A). Importantly, AAV9-*Acsl6* administration decreased E/A ratio, E/e' ratio, and left ventricle mass without affecting LVEF (Fig. 6D-H), demonstrating that *Acsl6* overexpression attenuates diastolic dysfunction in the progression of HFpEF. In addition, *Acsl6* overexpression increased exercise performance, improved glucose tolerance, and reduced plasma cholesterol (Fig. 6I-K). Mice with *Acsl6* overexpression exhibited increased heart mitochondrial function compared to controls (Fig. 6L-Q), suggesting that improved mitochondrial function may contribute to improved diastolic function. Taken together, these data demonstrate that enhancement of *Acsl6* levels in male mice improves mitochondrial function and helps prevent diastolic dysfunction in the HFpEF model.

Further, we overexpressed *Acsl6* and reproduced the HFpEF model in female mice using the same strategy. *Acsl6* expression was increased in the heart (Fig. 7A and Supplementary Fig. 11B). Body composition and food intake were not significantly affected by *Acsl6* overexpression after 7 weeks of HFD + l-NAME feeding (Fig. 7B and Supplementary Fig. 11C). Importantly, AAV9-*Acsl6* administration was sufficient to decrease heart weight, E/e' ratio, and left ventricle mass without affecting LVEF (Fig. 7C-H), demonstrating that *Acsl6* overexpression attenuates diastolic dysfunction in the progression of HFpEF. In addition, *Acsl6* overexpression increased exercise performance, improved glucose tolerance, and decreased plasma lipids, including plasma free fatty acids (Fig. 7I-L), suggesting that *Acsl6* helps maintain metabolic homeostasis in the HFpEF model. Mice with *Acsl6* overexpression exhibited decreased *Nppb* expression as compared with controls in the heart (Fig. 7M), indicating mitigated cardiac dysfunction. In contrast to male hearts, *Acsl6* did not significantly affect mitochondrial function in the female hearts (Fig. 7N-S), suggesting a sex-biased regulation of mitochondria. Taken together, these data demonstrate that enhancement of *Acsl6* levels improves diastolic dysfunction in both male and female mice through sex-biased mechanisms.

## Discussion

We have used systems genetics approaches that integrate genetics and omics technologies to understand mechanisms underlying HFpEF traits, including diastolic dysfunction. An aspect of particular interest in this study was the increased prevalence of HFpEF in women as compared to men. Using both mouse models and data from studies of heart failure in humans, we observed that males tend to have higher levels of mitochondrial gene expression (from both the nuclear and the mitochondrial genome) than females. In mice, we also found that mitochondrial DNA levels were higher in males and that the levels of both gene expression and mitochondrial DNA were regulated by male and female sex hormones. We further observed that mitochondrial DNA level was significantly associated with diastolic function, a key feature of HFpEF, in a panel of diverse strains. To further understand the role of mitochondrial function in HFpEF, we identified candidate causal genes for diastolic function based on the correlation between the *cis* component of gene expression and echocardiographic parameters reflecting diastolic function. Thus, the expression of a gene can be correlated with a trait either because the trait regulates gene expression (*trans*-regulation) or because the expression of the gene regulates the trait, the latter being consistent with *cis*-regulation. We chose the gene *Acsl6*, encoding a mitochondrial protein involved in lipid metabolism, for validation and follow-up. Each of these points is discussed in turn below.

HFpEF has become increasingly common with the increase in obesity and diabetes and now accounts for about half of all heart failure. The drugs that are effective in treating HfrEF have proven ineffective for HFpEF, and thus there has been an urgent need for a better understanding of the mechanisms underlying HFpEF and for the development of new drugs to treat it. Recent studies have shown that inhibitors of the sodium-glucose transporter 2 (SGLT2) that were developed for diabetes are also effective in reducing cardiovascular death and hospitalizations in patients with HFpEF[32]. Also, nicotinamide and drugs promoting NAD + synthesis appear promising based on studies in human and mouse models[33,34]. The interplay of metabolic disturbances and systemic inflammatory burden, named metainflammation, is considering a potential therapeutic target in HFpEF[35]. HFpEF appears to be very heterogenous and complex in etiology, with both metabolism and inflammation components[35,36]. This has made human studies difficult and mouse models that replicate many of the features of HFpEF have been developed[29]. Studies in the "two-hit" mouse model of HFpEF suggest that the Unfolded Protein Response (UPR) governs lipid accumulation and contractile performance[29,37]. Targeting UPR by Imeglimin prevents metabolic and cardiac abnormalities in the model[38].

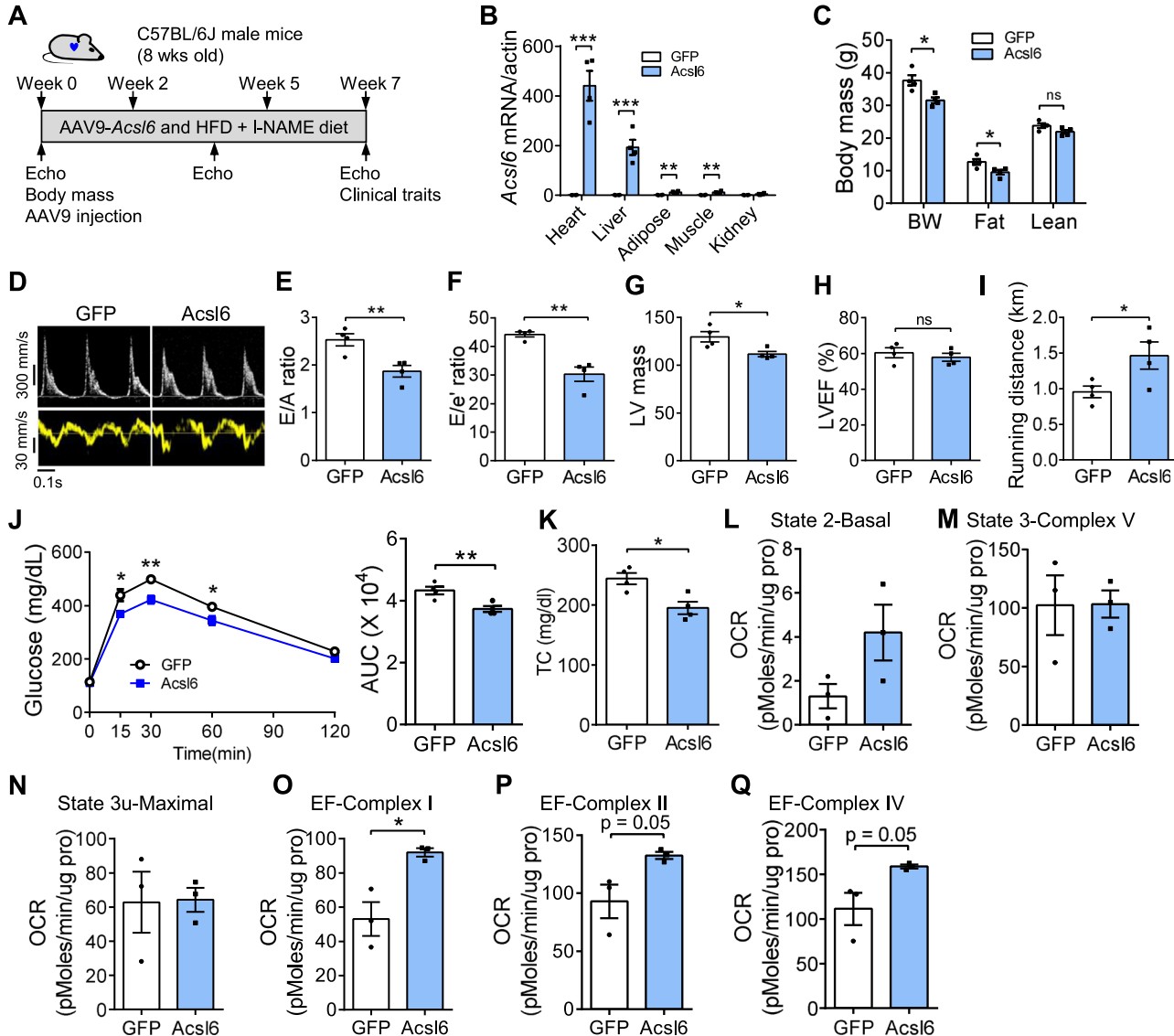

**Fig. 6 *Acsl6* improves diastolic function and mitochondrial function in C57BL/6 J male mice in HFpEF model.** C57BL/6 J male mice (8 weeks old) were injected with AAV9-GFP and AAV9-*Acsl6*. Mice were fed with HFD + l-NAME for 7 weeks to induce HFpEF. After feeding, diastolic function and mitochondrial function were examined. **A** Experimental timeline. **B** qRT-PCR showing *Acsl6* expression in the heart, liver, adipose tissue, skeletal muscle, and kidney after AAV injection. $N = 4$. p (heart) = 0.0003; p (liver) = 0.0008; p (adipose) = 0.002; p (muscle) = 0.004. **C** Body composition. BW body weight, Fat fat mass, Lean lean mass. $N = 4$. p (BW) = 0.015; p (fat) = 0.033. **D–H** Representative images of echo echocardiography (**D**), E/A ratio (**E**), E/e' ratio (**F**), LV mass (**G**), and LVEF (**H**) were examined. $N = 4$. **E** $p = 0.009$. **F** $p = 0.002$. **G** $p = 0.025$. **I–K** Running distance (**I**), glucose-tolerance test and area under curve (AUC, **J**), and plasma total cholesterol (**K**) were examined after the feeding. $N = 4$. **I** $p = 0.049$. **J** $p = 0.009$. **K** $p = 0.012$. **L–Q** OCR of isolated heart mitochondria were measured by Seahorse assay. The coupling assay measures basal respiration in presence of palmitoyl-carnitine, state 3 (ADP) and state 3 u (FCCP). The electron flow measures complex I (palmitoyl-carnitine), complex II (succinate), and complex IV (TMPD) respiration. $N = 3$. **O** $p = 0.018$. Each point represents a mouse. All data are presented as the mean ± SEM. ns not significant, *$P < 0.05$, **$P < 0.01$, and ***$P < 0.001$, by two-sided Student's *t*-test. Source data are provided as a Source Data file.

Among the characteristics of HFpEF is sex bias, as about two-thirds of patients are women, and women also tend to have more severe symptoms than males[9–12,39,40]. We note that the notion of female predilection has been called into question and may be due in part to the fact that women tend to be smaller than men[1,13–15]. One approach to examining sex differences is to study animal models where procedures such as gonadectomy can be applied and genetic and environmental factors can be controlled. We observed that in the "two-hit" HFpEF model female mice of multiple strains exhibited impaired diastolic function as compared to males. This contrasts with the findings of Tong et al.[21]. We suspect that the differences may be explained by the genetic

background since they used strain C57BL/6 N mice whereas we used strain C57BL/6 J mice as well as a number of other strains. Gene-by-sex interactions are difficult to detect in humans but are commonly observed in mice[22].

During the course of our studies of multiple mouse strains we found that males tend to have more mitochondrial DNA and increased expression of genes encoding mitochondrial proteins than do females (Fig. 2). Functional studies involving Seahorse analyses of oxygen consumption were consistent with these findings. We also observed both mitochondrial DNA levels and function were regulated by sex hormones, as testosterone increased these in gonadectomized males and estrogen decreased

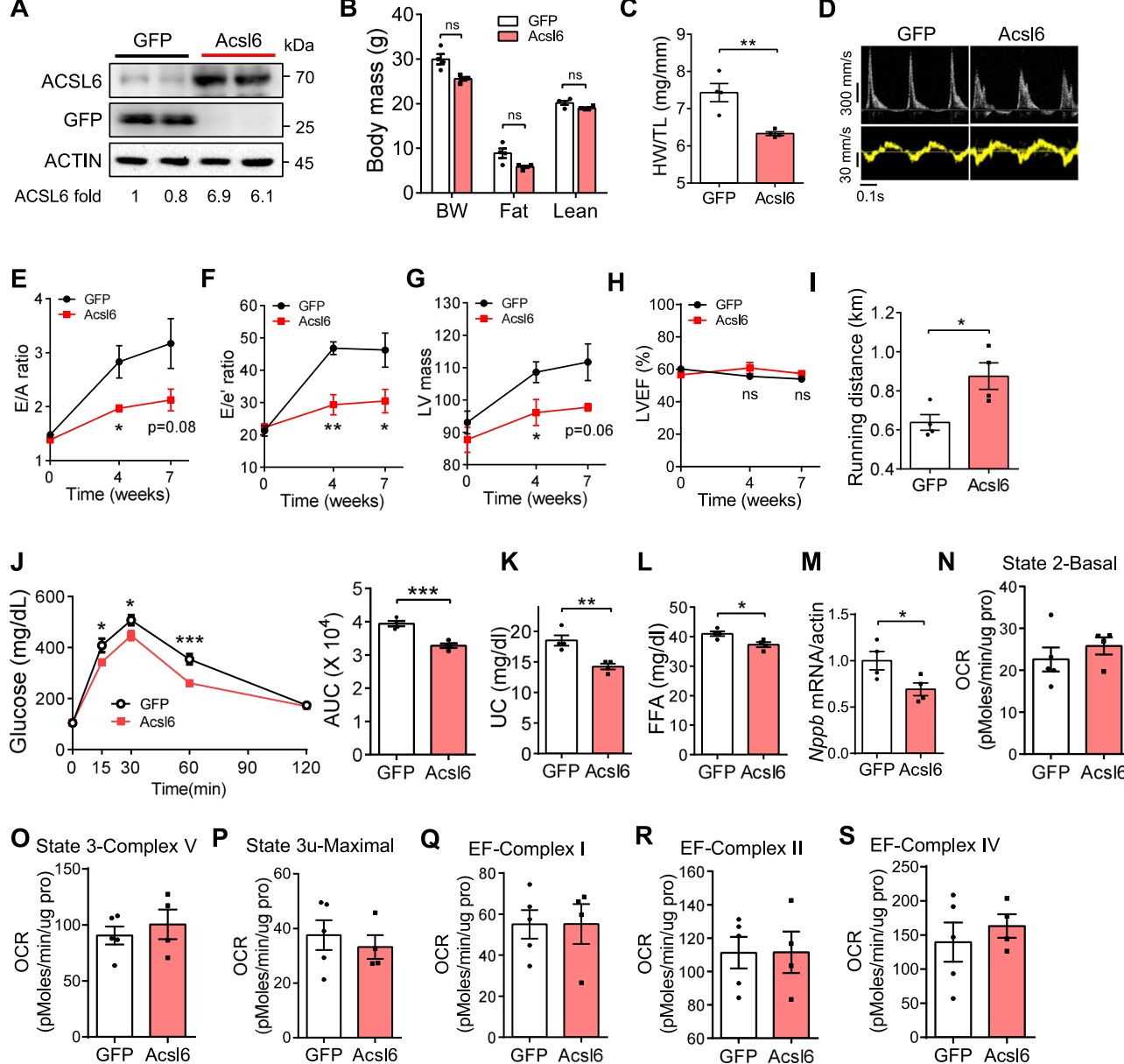

**Fig. 7 *Acsl6* improves diastolic function in C57BL/6 J female mice in HFpEF model.** C57BL/6 J female mice (8 weeks old) were injected with AAV9-GFP and AAV9-*Acsl6*. Mice were fed with HFD + l-NAME for 7 weeks to induce HFpEF. After feeding, diastolic function and mitochondrial function were examined. **A** Western blotting showing protein levels of ACSL6 in the heart of C57BL/6 J female mice treated with AAV9-GFP and AAV9-*Acsl6*. ACSL6-fold change was normalized to ACTIN. Experiment was independently repeated three times. **B** Body composition after the feeding. BW body weight, Fat fat mass, Lean lean mass. $N = 4$. **C–H** Heart wight/tibia length ratio (**C**), representative images of echocardiography (**D**), E/A ratio (**E**), E/e' ratio (**F**), LV mass (**G**), and LVEF (**H**) were examined. **C–F**, **H** $n = 4$. **G**, $n = 5$ (GFP) or 4 (Acsl6). **C** $p = 0.004$. **E** $p$ (week 4) = 0.03. **F** $p$ (week 4) = 0.003; $p$ (week 7) = 0.048. **G** $p$ (week 4) = 0.042. **I, J** Running distance and glucose-tolerance test (**J**) after HFD + l-NAME feeding. $N = 4$. **I** $p = 0.024$. **J** $p = 0.0007$. **K–M** Plasma unesterified cholesterol (**K**), free fatty acids (FFA, **L**), and heart *Nppb* mRNA level (**M**). $N = 4$. **K** $p = 0.005$. **L** $p = 0.019$. **M** $p = 0.043$. **N–S** OCR of isolated heart mitochondria were measured by Seahorse assay. GFP, $n = 5$; Acsl6, $n = 4$. Each point represents a mouse. All data are presented as the mean ± SEM. ns not significant, *$P < 0.05$, **$P < 0.01$, and ***$P < 0.001$, by two-sided Student's *t*-test. Source data are provided as a Source Data file.

these in gonadectomized females. We also observed that in human cohorts, the expression of most genes encoding mito-chondrial proteins was elevated in males as compared with females (Fig. 2), suggesting that humans have similar sex differences in heart mitochondrial function. In mice, cardiomyocytes, rather than other cell types account for the differences in mito-chondrial function.

We suspected that the mitochondrial differences could be responsible in part for the sex differences in HFpEF prevalence since a number of studies have provided convincing evidence of

the role of mitochondria in heart failure[23,41,42]. Supporting this idea were several findings: First, in a genetic study of isoproterenol-induced heart failure in mice we found that genes associated with diastolic dysfunction were most strongly enriched in mitochondrial pathways (Fig. 1H). Second, in the isoproterenol study mitochondrial copy number was correlated with several measures of diastolic dysfunction. Third, gonadectomy studies in the "two-hit" HFpEF model decreased mitochondrial function and also worsened diastolic function in males whereas it increased mitochondrial function and improved diastolic

function in females (Fig. 3). Fourth, the studies of *Acsl6* provide evidence for a causal connection.

The *Acsl6* gene, encoding a mitochondrial protein involved in fatty-acid metabolism, was validated by overexpression in the heart using an adeno-associated viral vector. Overexpression improved diastolic function, reduced left ventricular mass, increased running distance, and improved glucose disposal (Figs. 6 and 7). It also improved aspects of mitochondrial function, as judged by the oxygen consumption rate of isolated mitochondria in males but not females (Figs. 6 and 7). One limitation of our study is that as *Acsl6* shows some overexpression in other tissues, notably in the liver, additional studies will be required to determine its mechanisms of action. The mechanistic link between *Acsl6* and diastolic function may relate to ATP production. The heart requires a tremendous amount of ATP during relaxation as well as contraction, as calcium needs to be sequestered in the endoplasmic reticulum.

The finding that sex hormones regulate mitochondrial levels may have clinical implications. The levels of testosterone decline in elderly men and levels of estrogen decrease following menopause in women[23,43]. Testosterone treatment, however, has diverse effects that complicate its usefulness as a therapeutic agent. Also, a meta-analysis of eight small studies of heart failure with a total of 170 subjects failed to observe significant improvement following testosterone treatment[43].

## Methods

**Mice**. All animal experiments were approved by the University of California Los Angeles (UCLA) Animal Care and Use Committee, in accordance with Public Health Service guidelines. All mice were purchased from the Jackson Laboratory. Mice were maintained on a 12 h light/dark cycle from 6 am to 6 pm at ambient temperature (~72 F) with controlled humidity (~45%) in pathogen-free conditions. Animals were randomly assigned to each treatment group and experiments were performed under standard laboratory procedures of randomization and blinding. No wild animals or field-collected samples were used in the study.

**HMDP models**. HFpEF was induced by HFD + l-NAME feeding for 7 weeks[29]. Eight-week-old mice were fed with a chow diet (Teklad, Cat# 2916) or HFD (Research Diets # D12492). Nω-Nitro-L-arginine methyl ester hydrochloride (l-NAME) (Sigma # N5751-25G) was dissolved in drinking water (0.5 g/L, pH = 7.4). Echocardiography was performed before and after the diet to assess cardiac function. For ISO-HMDP, 9-week-old female mice from 105 inbred strains were treated with isoproterenol (30 mg/kg/day) via an intra-abdominally implanted osmotic pump for 21 days. Mice were maintained on a chow diet and echocardiography was performed weekly to assess cardiac function. Body composition was determined biweekly using nuclear magnetic resonance (NMR, Bruker Minispec). Upon sacrifice, tissues were weighed and instantly frozen in liquid nitrogen. Blood was collected in a BD Microtainer (Tubes with K₂EDTA, # 365974) and plasma was collected after $9300 \times g$ for 5 min at 4 °C.

**Gonadectomy and hormone replacement**. Gonadectomy was performed as previously described[44]. Briefly, at 6 weeks of age the mice were subjected to gonadectomy, with mice treated with sham and hormone pellet implantation after surgery serving as controls. Mice were gonadectomized under isoflurane anesthesia (1–2% isoflurane in 95% oxygen). Scrotal skin of male mice was bilaterally incised, testes removed, and the incisions closed with suture and wound clips. Ovaries of female mice were removed through an incision below the ribcage (about midway between the hip and the bottom of the ribcage, and about 2/3 of the way from ventrum to dorsum). In sham-operated control mice, incisions were made and closed as described above but the gonads remained intact. For hormone pellet implantation after gonadectomy, a 5 mg pellet of 5a-dihydrotestosterone 90-day release (Innovative Research of America, FL) and 0.5 mg estradiol 90-day release were used with tubing, respectively. Carprofen (Rimadyl, 6 mg/kg body weight) was injected for three consecutive days after the surgery.

**Trans-thoracic echocardiography**. The mice were anesthetized and maintained with 1–2% isoflurane in 95% oxygen. *Trans*-thoracic echocardiography was conducted with Vevo 2100 high-frequency, high-resolution digital imaging system (VisualSonics) equipped with a MS400 MicroScan Transducer. A parasternal short axis view was used to obtain M-mode images for analysis of fractional shortening, ejection fraction, and other cardiac functional parameters. The apical four-chamber view was used to obtain tissue Doppler imaging (TDI) mode and Pulse-wave Doppler (PWD) mode for analysis of myocardial velocity and blood flow velocity,

respectively. Echocardiographic results in the different groups of mice were listed in Supplementary Table 1.

**Exercise exhaustion test**. After three days of acclimatization to a treadmill, an exhaustion test was performed on the mice. Mice ran on the treadmill (20°) at the warming-up speed of 5 m/min and the speed was increased gradually to 18 m/min. The speed was kept at 18 m/min until the mice were exhausted. Exhaustion was defined as the inability of the mice to return to running within 10 s of direct contact with an electric-stimulus grid. Running time was recorded and running distance was calculated. Anesthesia was not required and mice were returned to their original cages immediately after the test.

**Food intake**. After 7 weeks of HFD + l-NAME feeding, daily food intake was calculated by subtracting the remaining food from the initially premeasured food for 3 consecutive days.

**Mitochondrial isolation and function**. Real-time measurement of oxygen consumption rate (OCR) in isolated mitochondria was carried out using the XF24 Extracellular Flux Analyzer (Agilent)[45]. Upon sacrifice, the heart was instantly washed in cold PBS and homogenized in MSHE buffer. Mitochondria were isolated by dual centrifugation ($800 \times g$ and $8000 \times g$) and resuspended in MAS buffer. For respirometry assays, 6 µg of protein were loaded per well. The coupling assay measurements were obtained in the presence of 40 µM palmitoyl-carnitine and 100 µM malate, and after the sequential injection of 4 mM ADP (State 3), 2.5 µM oligomycin (ATP synthase inhibitor, which allows determination of ATP-linked respiration), 4 µM FCCP (an uncoupler, which allows determination of maximal mitochondrial respiratory capacity), and 1.5 µg/ml Antimycin A (inhibitors of complex III of the electron transport chain, which allows determination of uncoupled respiration). The electron flow assay measured complex I (40 µM palmitoyl-carnitine and 100 µM malate in presence of 4 µM FCCP), complex II (10 mM succinate), and complex IV (100 µM TMPD) respiration. OCR was normalized per protein content using a Bradford assay.

**Adult cardiomyocyte isolation and cellular bioenergetics**. Adult mouse cardiomyocytes were isolated using a Langendorff system as previously described[46,47] with some modifications to allow cardiomyocyte plating. Mice were injected with 100 USP unit heparin (Sagent, 25021-400-10) and then anesthetized with 100 µL pentobarbital (Vortech Pharmaceuticals). Hearts with intact aorta were removed, and lungs were cleared from cardiac tissue. The aorta was cannulated and the hearts perfused with Tyrode's solution (13 mM NaCl, 0.54 mM KCl, 0.06 mM NaH₂PO₄, 0.1 mM MgCl₂, 1 mM HEPES, 10 mM glucose, 0.5 mM EGTA, and 10 mM BDM, pH 7.37) oxygenated with 95% (v/v) O₂–5% (v/v) CO₂) at 37 °C) for 20 min. Cardiac tissue was then perfused with Digestion Buffer (5 mg Protease (Sigma) and 50 mg Type 2 Collagenase (Worthington) in Tyrode's solution) for 10–15 min. After digestion, the organs were finally perfused with KB Buffer (130 mM NaCl, 25 mM KCl, 10 mM KH₂PO₄, 2 mM MgSO₄, 10 mM glucose, 20 mM taurine, 5 mM creatine, 100 mM potassium glutamate, 10 mM aspartic acid, 10 mM HEPES, 50 µM CaCl₂, pH 7.18, oxygenated with 95% O₂–5% (v/v) CO2.) for 5 min. Hearts were removed from the Langendorff apparatus and placed in Dissociation buffer (10% FBS and 25 µM Blebbistatin in KB buffer). Atria and right ventricle were removed, left ventricular myocytes were dissociated into solution, filtered (100 µm strainer), and calcium was gradually introduced in increasing increments (250, 500, and 1000 µM) before adding culture media (MEM supplemented with 25 mM HEPES, 10% FBS, penn/strep, and 25 µM blebbistatin). Cardiomyocytes were seeded in an XF24 plate (precoated with 5 µg/mL laminin, Sigma) at 100,000 cells per well. One hour after plating, unattached myocytes were washed with culture media. Oxygen consumption rate (OCR) was measured before and after the sequential injection of 1.5 µM oligomycin, 1.5 µM FCCP, and 4 µM of rotenone/myxothiazol. Mixing, waiting, and measurement times were 3, 2, and 3 min, respectively. Measures were normalized by total protein by Bradford (Thermo Fisher).

**Adeno-associated virus (serotype 9)**. AAV9-cTnT-m-Acsl6-WPRE and AAV9-cTnT-m-eGFP-WPRE were purchased from Vector Biosystems Inc (AAV-251968). AAV stock was purified in PBS with 5% glycerol and diluted with sterile saline before tail vein injection. Each mouse received 2e12 genome copies virus in 100 uL total volume.

**RNA extraction and reverse transcription**. Upon sacrifice, tissues were instantly frozen in liquid nitrogen. Cells or frozen tissue were homogenized in QIAzol Lysis Reagent (Qiagen, 1 mL per 20 mg tissue), and following chloroform (200 µL per 1 mL QIAzol) phase separation RNA extraction was carried out using miRNeasy Mini Kit (Qiagen # 217004) as recommended by the manufacturer. High-quality RNA was eluted in ~50 µL RNase-free water and assessed for purity and concentration using a Nanodrop ND-100 Spectrophotometer. Two micrograms of total RNA per sample were reverse transcribed using a MultiScribe Reverse Transcriptase kit (Applied Biosystems # 4311235) with random primers. Reverse-transcribed cDNA was then diluted in water (10–20× dilution) for qPCR analysis.

**Quantitative PCR**. Quantitative PCR was carried out using a PowerTrack SYBR Green Master Mix (100029284) as recommended by the manufacturer. Samples were run on a LightCycler 480 II (Roche) and analyzed with the Roche LightCycler 1.5.0 Software. qPCR targets were normalized to actin and quantified using the delta Ct method. All qPCR primer sequences were obtained from PrimerBank (https://pga.mgh.harvard.edu/primerbank/). All Sequences to qPCR primers are listed below:

| Species | Gene | Sequence (5'−3') |
|---------|------|------------------|
| Mouse | Actin-F | GGCTGTATTCCCCTCCATCG |
| Mouse | Actin-R | CCAGTTGGTAACAATGCCATGT |
| Mouse | Fabp3-F | ACCTGGAAGCTAGTGGACAG |
| Mouse | Fabp3-R | TGATGGTAGTAGGCTTGGTCAT |
| Mouse | Acsl6-F | GCACAGAGCAGTTCGTTGGT |
| Mouse | Acsl6-R | ATGTAGCTGATAGACCCTGGG |
| Mouse | Fasn-F | GGAGGTGGTGATAGCCGGTAT |
| Mouse | Fasn-R | TGGGTAATCCATAGAGCCCAG |
| Mouse | Acaca-F | ATGGGCGGAATGGTCTCTTTC |
| Mouse | Acaca-R | TGGGGACCTTGTCTTCATCAT |
| Mouse | Nppb-F | GAGGTCACTCCTATCCTCTGG |
| Mouse | Nppb-R | GCCATTTCCTCCGACTTTTCTC |

**Western blotting**. Heart tissues were lysed in Whole Cell Extraction buffer (WCE) containing 62.5 mM Tris-HCl (pH 6.8), 2% (wt/v) Sodium dodecylsulfate, and protease and phosphatase inhibitor (Fisher # 78441). Protein content was measured using BCA protein assay kit (Pierce) and samples were denatured in 4× LDS loading buffer (Life Technologies) at 99 °C for 5 min. Samples were then loaded into 4–12% Bis-Tris gels (Invitrogen) and separated out at 80 volts. Protein was then transferred to PVDF membranes (Immobilon) for 2 h at 100 volts. Following transfer, membranes were washed with TBST, and then blocked in 5% milk (Gibco) in TBST for 1 h at room temperature. Membranes were then placed in primary antibodies on a shaker overnight at 4 °C. Primary antibodies were used as follows: rabbit ACSL6 (Thermo Fisher # PA5-30465, 1:1000), rabbit ACTIN (Cell Signaling # 4967 S, 1:2000), mouse GFP (Cell Signaling # 2955 S, 1:2000). The following day, membranes were washed in TBST and then placed in secondary antibodies (Cell Signaling, anti-rabbit IgG, HRP-linked antibody, Cat# 7074P2,1:10,000; anti-mouse IgG HRP-linked antibody, Cat# 7076 S, 1:10,000) for 1.5 h at room temperature. Blots were then washed in TBST and placed in Amersham ECL detection solution (GE health sciences). Blots were imaged using Amersham ImageQuant™ 800 Western Blot Imaging Systems (Cytiva) and bands were quantified using ImageJ Software. Source data are provided as a Source Data file.

**RNA library preparation and sequencing**. Upon RNA extraction with miRNeasy Mini Kit, total RNA quality was validated with BioANAlyzer (all samples had RIN > 8). RNA libraries were prepared by the sequencing facility at the UCLA Technology Center for Genomics & Bioinformatics (TCGB) using Nugen Universal mRNA-Seq kit. Gene expression in the heart from the HMDP strains was analyzed using 2 ×50 NovaSeq S2 (~37.5 million reads per sample). Reads were quantified against the GRCm38.p6 mouse reference transcriptome (Ensembl release 97) using kallisto version 0.46.0. Differential expression analysis was performed with DESeq2 1.28.1.

**Intraperitoneal glucose-tolerance test**. Intraperitoneal glucose-tolerance tests (i.p. GTT) were performed by injection of glucose (2 g/kg body weight in sterile saline) after 16 h fasting. Tail blood glucose levels (mg/dL) were measured with a glucometer before (0 min) and at 15, 30, 60, and 120 min after glucose administration.

**Plasma lipid measurement**. Plasma lipids were measured as described previously[48]. Briefly, mice were fasted for 4 h prior to retroorbital bleeding using isoflurane. Plasma cholesterol concentrations were determined by enzymatic procedures in 96-well plates using a Biomek 1000 Automated Laboratory Workstation (Beckman). Free fatty acids were measured with Wako Diagnostics HR series NEFA-HR (2) kit. Samples were measured at a wavelength of 490 nm with a Vmax Microplate Reader (Molecular Devices, Inc.). Each sample was measured in triplicate.

**Mitochondrial DNA content**. Mitochondrial DNA content was measured as described previously[25,26] with modifications. About 20 mg heart tissue was digested with 500 µL digestion buffer (100 mM NaCl, 10 mM Tris-Cl, pH 8, 25 mM EDTA, pH 8, 0.5% SDS, 0.2 mg/ml proteinase K) at 55 °C overnight (5–18 h until the sample is clear). After digestion, 500 µL phenol/chloroform/isoamyl alcohol (25:24:1 ratio, prepared fresh) was added to the DNA. After 1 min of vortex, the samples were centrifuged at 9300 × $g$ for 10 min at room temperature. The top phase containing total DNA (mitochondrial and nuclear) was carefully transferred to a new tube. An equal volume (~400 µL) of CI (chloroform: isoamyl alcohol 24:1)

was added, mixed gently for 1 min, and centrifuged for 10 min at 10,000. The top phase containing DNA was carefully transferred to a new tube. Then 200 µL of 7.5 M ammonium acetate and 2–2.5 volumes of ice-cold 100% ethanol were added to DNA, mixed gently, and placed at −70 °C for 20–30 min. The mix was then centrifuged in a fixed-angle microcentrifuge at maximum speed for 15 min. After removing the supernatant, 1 mL of 75% ethanol was added. The tube was inverted gently and speed at maximum speed for 15 min. Then the supernatant was removed and the DNA pellet was dried at room temperature. The pellet was dissolved in 200 µL TE buffer. Both mitochondrial and nuclear DNA were amplified by quantitative PCR with 25 ng of total DNA using primers in the D-loop region and Tert gene, respectively. Mitochondrial DNA content was normalized to nuclear DNA, and was calculated using the equation 2 × 2DCt (DCt = D-loop Ct − Tert Ct).

Primer sequences for mtDNA:
Forward 5'-AATCTACCATCCTCCGTGAAACC-3';
Reverse: 5'-TCAGTTTAGCTACCCCCAAGTTTAA-3'.
Tert control:
Forward 5'-CTAGCTCATGTGTCAAGACCCTCTT-3';
Reverse 5'-GCCAGCACGTTTCTCTCGTT-3'.

**Expression quantitative trait loci (eQTL) analysis and systems genetics approach**. Genome-wide association study (GWAS) for gene expression was performed with FaST-LMM (fast spectrally transformed linear mixed models). eQTLs with peak single nucleotide polymorphism (SNP) mapped within 1 Mb of gene position ($p < 1.4e-3$) are referred to as cis-eQTLs. The systems genetics approach to identify candidate causal genes for complex traits was performed as previously described[31]. Briefly, we leveraged gene expression variation due to local or cis-eQTL SNP genotypes to identify candidate causal genes. Thus, across the mouse diversity panel, there was a significant correlation between the cis component of Acsl6 gene regulation and the E/A ratio. Such correlations are consistent with a causal role for the gene since the cis variation is controlled by upstream local genetic differences rather than in trans by the downstream trait.

**Statistical analysis**. All computational procedures were carried out using R statistical software. Data were analyzed with GraphPad software (2018 GraphPad Software, Inc, USA). Pathway analysis was performed with The Database for Annotation, Visualization, and Integrated Discovery (DAVID) https://david.ncifcrf.gov/home.jsp. Correlations and associated $p$-values were calculated with the biweight midcorrelation, which is robust to outliers and associated p-value. Single comparisons between two groups were performed using two-sided Student's $t$-tests with 95% confidence intervals. Comparisons involving multiple time points were assessed using an ordinary one-way ANOVA followed by Tukey's multiple comparisons test or using a two-way ANOVA followed by Sidak's multiple comparisons test. All data presented as mean ± SEM. Values were considered significant at $p < 0.05$. *$p < 0.05$, **$p < 0.01$, ***$p < 0.001$, and ****$p < 0.0001$.

## Data availability

The data supporting the findings from this study are available within the manuscript and its supplementary information. The RNA-sequencing data were deposited to the Gene Expression Omnibus (GEO) database (https://www.ncbi.nlm.nih.gov/geo/) under the accession number GSE194198 (HF/HS heart HMDP RNA-seq data) and GSE194151 (HFpEF heart RNA-seq data). Genome Reference Consortium Mouse Build 38 patch release 6 (GRCm38.p6) is released from Ensembl (Ensembl 97: http://jul2019.archive.ensembl.org/Mus_musculus/Info/Index). Source data are provided with this paper.

## Code availability

The R script used to perform the heatmap, volcano plot, PCA plot, Manhattan plot, trait–trait correlation, gene–trait correlation, and cis-eQTL is available at https://github.com/yangcao208/sex-differences.

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

## Acknowledgements
We would like to acknowledge the undergraduate students Gillian Campbell, Kristina Guardino, Chris Tai, and Christopher Li for assistance with mouse experiments and RNA extraction. We thank Rosa Chen for help in preparing the manuscript. This work was supported by NIH grants DK120342 and HL147883, as well as DOD grant W81XWH2110115. Karthickeyan Chella Krishnan was supported by NIH grant R00DK120875.

## Author contributions
Y.C. and A.J.L. designed the experiments. Y.C., L.V., Y.C.W., K.C.K., M.R.G., T.H.K., Z.Z., and S.C. performed the experiments. Y.C., C.P., and L.V. analyzed raw data. L.V., T.M.M., M.M.S., J.W., Y.W., T.M.V., K.R., and A.J.L. reviewed the data and made substantial contributions to improving the studies. Y.C. and A.J.L. wrote the manuscript, which was reviewed by all authors.

## Competing interests
The authors declare no competing interests.
