## [Peer Review File · Nature Communications]

REVIEWER COMMENTS

Reviewer #1 (Remarks to the Author):

The study comprises a series of experiments that investigate sex differences in heart mitochondria and their relationship to diastolic dysfunction. The study delineates the *Acsl6* gene as a regulator of diastolic dysfunction in the context of HFpEF in female mice.

Major points:

1. The authors should provide more clarity about the overlap of the disease QTL on chr11 with the cis-eQTL for *Acsl6*. Please provide this as an integrated figure showing the overlapping chromosomal region etc. Also, please provide the outcome of cis-eQTL analysis of other genes in the interval. Provide a clearer reasoning for why *Acsl6* was considered the causal gene.
2. In the Highlights the authors state: 'Systems genetic analyses of heart transcriptomes in the HMDP as well as human cohorts identified the mitochondrial fatty acid metabolism gene, *Acsl6*, as a sex-specific regulator of diastolic dysfunction in the context of HFpEF.' Can the authors explain on the basis of which experiments is the claim of 'sex-specific regulation'? To the understanding of this reviewer such a statement can only be made when for instance the authors demonstrate an effect of *Acsl6* in one sex but not the other.
3. The Reviewer has the impression that the data presented in Figure 2, D-H, may be driven by some outliers. The authors state that 'Correlations and associated p values were calculated with the biweight midcorrelation, which is robust to outliers and associated p-value.' Has a sensitivity analysis been performed by re-analysing without the samples at extreme high mtDNA content?

Minor comment

Measurement of mt DNA content is central to the manuscript and deserves a brief explanation of how this was done (not only referring to previous papers).

Page 6: 'We also sought to identify sex-specific regulators of diastolic function by integrating the ISO HMDP and "two-hit" HFpEF cohorts, as well as the human heart failure cohort.' It is unclear what is meant by this and where the emerging data is presented. Figure S3A in the sentence thereafter concerns only human heart data.

Page 6: 'In particular, we focused on a genome-wide association locus for E/A ratio on mouse chromosome 8..' I suppose you mean genome-wide significant?

Figure 5: The arrow in panel F points to chr11 whereas in the text on page 6 the authors talk of chr8 (line 250). The mouse *Acsf6* gene is indeed on chr11. Please correct text.

Page 6: 'Gonadectomized mice exhibited reduced *Acsf6* expression in the heart after HFpEF (Fig. 5D-5E), suggesting its contribution to increased mitochondrial and metabolic dysfunction in HFpEF mice.' Please clarify in text that these were male mice.

In the last paragraph of the Results one should mention earlier in the text that studies were done in female mice.

One only discovers in the legend of Figure 5 that the eQTL analysis of *Acsf6* was conducted in female mice after 3 weeks of isoproterenol infusion (ISO-HMDP). Such info should be added to the Results. Please check ms throughout for similar instances.

The authors refer to *Acsf6* and other genes as mitochondrial genes whereas they mean genes expressed in mitochondria? While this term may be commonly used it can be confusing in this paper where mitochondrial copy number is also investigated.

The role of *Acsf6* can be better connected to effects of lower mitochondrial levels in the Discussion.

Reviewer #2 (Remarks to the Author):

The manuscript examines the basis for sex differences in HFpEF by harnessing the power of the Hybrid Diversity Mouse Panel. They zeroed in on mitochondrial differences between male and female heart, performed gonadectomies, measured mitochondrial function and identified a protective gene. The general conclusions were that male hearts have more mitochondria and more mitochondrial gene expression than females and that this difference is driven hormonally. These results are important and may be relevant to humans with HFpEF.

I have several comment/suggestions:

1. Since mito DNA is higher in male versus female hearts, if they normalize their mito gene expression data to mito DNA copies, what is seen?

2. The differences between this group's results and those of the Hill lab in their two hit model of HFpEF seem very important as the field has been searching for a HFpEF model in which females show worse phenotypes than males is really important to "get out there" since the Hill lab did not see the expected female worse female difference. But, If background genetic strain is the explanation, that flies in the face of the high genetic diversity in the human clinical population of HFpEF. So, this aspect needs to be discussed.

3. The authors talk about mitochondrial function in their results from the beginning, but they are using things other than function as surrogate for function. It isn't until Figure 4 that they actually measure function.

4. In Figure 5 E, the author should state in the text that they only did gonadectomies in males. Why did they not do them in females as they did throughout the paper.

Reviewer #3 (Remarks to the Author):

These authors report that levels of "heart mitochondria" are higher in males than in females in both humans and mice. Mitochondrial DNA copy number and levels of mitochondrial genes are increased by testosterone and decreased by estrogen. The paper goes on to make some inferences regarding myocardial diastolic dysfunction.

These authors have previously reported differences in mitochondrial biomass based on sex in white adipose tissue and liver, so the report here extending this finding to heart is at some level incremental. It is not a "surprising discovery" (first sentence of Discussion).

The authors often-stated assertion that HFpEF is more common in females is based on old and now outdated literature. Please see Ho et al, *Circ Heart Fail.* 2013; Brouwers et al, *Eur Heart J* 2013; Dunlay et al, *Nature Reviews* 2017; Pandey et al, *Circ* 2018. The notion of a female predilection to HFpEF has largely been abandoned, or at the very least called into question. Furthermore, even if there were a female predisposition for HFpEF, there is no evidence for a cellular/molecular basis; rather, the fact that females tend to be smaller than males might play a role.

I do not understand the leap from mitochondrial biomass to diastolic dysfunction. How do the authors suggest that they are linked mechanistically? Furthermore, HFpEF is marked by a phenotype with numerous features well beyond just diastolic dysfunction. Indeed, throughout the paper the authors seem to equate diastolic dysfunction with HFpEF which is unequivocally wrong.

The authors argue that their studies of “heart mitochondria” reflect cardiomyocyte mitochondria given that mitochondrial biomass in these cells is quite high. That said, the observation remains indirect. The findings should be corroborated in isolated cardiac myocytes.

The data to correlate mitochondrial biomass with cardiac phenotypes are undoubtedly driven by outlier data points, and hence they are unconvincing (Figs 2D-2H).

To assert that *Acsf6* levels are “dramatically lower” in females based on data in Figure 5B is a huge over-statement. And to infer from these data that *Acsf6* may be a “causal driver” is yet another stretch.

The manuscript is laid out in an atypical manner, including presentation of the study findings in the Introduction rather than using that section of the paper to lay out the knowns and unknowns and the questions to be addressed in the study. Also, the text is quite redundant in several places. The Discussion is largely a recapitulation of the Results.

On page 5, the authors cite a paper reporting that female mice are less susceptible to HFpEF than males, but the paper reporting this is not cited.

The term “female mitochondria” is odd and should be avoided.

Response to reviewers' comments:

Reviewer #1 (Remarks to the Author):

The study comprises a series of experiments that investigate sex differences in heart mitochondria and their relationship to diastolic dysfunction. The study delineates the *Acs16* gene as a regulator of diastolic dysfunction in the context of HFpEF in female mice.

Major points:

1. The authors should provide more clarity about the overlap of the disease QTL on chr11 with the cis-eQTL for *Acs16*. Please provide this as an integrated figure showing the overlapping chromosomal region etc. Also, please provide the outcome of cis-eQTL analysis of other genes in the interval. Provide a clearer reasoning for why *Acs16* was considered the causal gene.

Response: We have now added a locus zoom plot of the region of chromosome 11 associated with E/A ratio (Fig. S7). However, *Acs16* was not identified based on overlap with the GWAS of E/A ratio and indeed the GWAS locus for E/A ratio was about 30 Mb from the *Acs16* gene. The locus was quite broad and the peak region contained over 30 genes (Fig. S7). Rather, we used a systems genetics strategy which leverages gene expression variation due to local or *cis*-eQTL SNP genotypes to identify candidate causal genes (Fig. 5F-5I and S8). Thus, across the mouse diversity panel there was significant correlation between the *cis* component of *Acs16* gene regulation and the E/A ratio. Such correlations are consistent with a causal role for the gene, since the *cis* variation is controlled by upstream local genetic differences rather than by the downstream trait. We then validate the causality using *Acs16* overexpression in a mouse model of HFpEF. We have used this approach to identify candidate causal genes for several complex traits that were subsequently validated. The approach is outlined in Gusev et al. *Nature Genetics*, 2016¹ and Seldin et al, *Nature Metabolism*, 2019².

We apologize for the confusion and have more clearly outlined our approach in the revised Methods (page 15): “The systems genetics approach to identify candidate causal genes for complex traits was performed as previously described¹. Briefly, we leveraged gene expression variation due to local or *cis*-eQTL SNP genotypes to identify candidate causal genes. Thus, across the mouse diversity panel there was significant correlation between the *cis* component of *Acs16* gene regulation and the E/A ratio. Such correlations are consistent with a causal role for the gene, since the *cis* variation is controlled by upstream local genetic differences rather than in trans by the downstream trait.”

In addition to the sex-biased gene expression and gene-trait correlation in the ISO-HMDP (Fig. 2B-2C and Fig. 5G-5I), we also screened the “two-hit” HFpEF cohorts to identify sex-biased regulators in response to the HFpEF diet (Fig. S5) and examined the enriched pathways of sex-biased genes in human populations (Fig. S6A). *Acs16* was identified as a candidate that was significantly correlated with E/A ratio and exhibited sex-biased expression in response to HFpEF. We have now attempted to clarify this point (page 6): “We also sought to identify sex-biased

regulators of diastolic function by overlapping the gene expression data in female ISO HMDP (Fig. 2B-2C) and “two-hit” HFpEF cohorts (Fig. S5), as well as the human heart failure cohort (Fig. 2D-2G)³. In the human heart failure cohort, sex-biased DEGs were enriched in pathways including cell differentiation and mitochondrion (Fig. S6A). Among the sex-biased genes, *Acs16* (Acyl-CoA Synthetase Long Chain Family Member 6) expression was significantly lower in female hearts compared with males across the HMDP (Fig. 5B) and was further decreased after HFpEF development (Fig. 5C and 5D).”

2. In the Highlights the authors state: ‘Systems genetic analyses of heart transcriptomes in the HMDP as well as human cohorts identified the mitochondrial fatty acid metabolism gene, *Acs16*, as a sex-specific regulator of diastolic dysfunction in the context of HFpEF.’ Can the authors explain on the basis of which experiments is the claim of ‘sex-specific regulation’? To the understanding of this reviewer such a statement can only be made when for instance the authors demonstrate an effect of *Acs16* in one sex but not the other.

Response: We agree with the reviewer and have now substituted the term “sex-biased” to clearly indicate that the sex differences were quantitative rather than qualitative.

3. The Reviewer has the impression that the data presented in Figure 2, D-H, may be driven by some outliers. The authors state that ‘Correlations and associated p values were calculated with the biweight midcorrelation, which is robust to outliers and associated p-value.’ Has a sensitivity analysis been performed by re-analysing without the samples at extreme high mtDNA content?

Response: The correlations were performed using biweight midcorrelation (bicor), which is a robust alternative to Pearson correlation that is not as affected by outliers because the underlying calculations are based on medians and not means. None of the samples were considered outliers when using median absolute deviations (a robust alternative to Z-scores). In addition, none of the Z-scores exceeded the maximum theoretical Z-scores for this sample size (<https://cran.r-project.org/web/packages/outliers/index.html>).

As suggested by the reviewer, we have also now reanalyzed the data following removal of samples at the extreme high mtDNA content (Fig. S2). In most cases the correlations remained statistically significant or suggestive. Notably, we have corrected the original correlations by filtering out the control mice from the isoproterenol treated mice. The updated plots are shown in Fig. 1I-1L and S2.

Minor comment

Measurement of mt DNA content is central to the manuscript and deserves a brief explanation of how this was done (not only referring to previous papers).

Response: Thank you for pointing this out. We have included a detailed explanation in Methods (page 14): “About 20 mg heart tissue was digested with 500 µL digestion buffer (100 mM NaCl,

10 mM Tris-Cl, pH 8, 25 mM EDTA, pH 8, 0.5% SDS, 0.2 mg/ml proteinase K) at 55°C overnight (5-18h until the sample is clear). After digestion, 500 µL phenol/chloroform/isoamyl alcohol (25:24:1 ratio, prepared fresh) was added to the DNA. After 1 min of vortex, the samples were centrifuged at 10,000 rpm for 10 min at room temperature. The top phase containing total DNA (mitochondrial and nuclear) was carefully transferred to a new tube. An equal volume (~400 µL) of CI (chloroform: isoamyl alcohol 24:1) was added, mixed gently for 1 min and centrifuge for 10 min at 10,000. The top phase containing DNA was carefully transferred to a new tube. Then 200 µL of 7.5 M ammonium acetate and 2 to 2.5 volumes of ice-cold 100% ethanol were added to DNA, mixed gently and placed at -70°C for 20-30 min. The mix was then centrifuged in a fixed-angle microcentrifuge at maximum speed for 15 min. After removing the supernatant, 1 mL of 75% ethanol was added. The tube was inverted gently and speed at maximum speed for 15 min. Then the supernatant was removed and the DNA pellet was dried at room temperature. The pellet was dissolved in 200 µL TE buffer. Both mitochondrial and nuclear DNA were amplified by quantitative PCR with 25 ng of total DNA using primers in the D-loop region and Tert gene, respectively. Mitochondrial DNA content was normalized to nuclear DNA, and was calculated using the equation $2 \times 2DCt$ ($DCt = D\text{-loop } Ct - Tert \text{ } Ct$). Primer sequences for mtDNA: Forward 5'-AATCTACCATCCTCCGTGAAACC-3'; Reverse: 5'-TCAGTTTAGCTACCCCAAGTTTAA-3'. Tert control: Forward 5'-CTAGCTCATGTGTCAAGACCCTCTT-3'; Reverse 5'-GCCAGCACGTTTCTCTCGTT-3'."

Page 6: ‘We also sought to identify sex-specific regulators of diastolic function by integrating the ISO HMDP and “two-hit” HFpEF cohorts, as well as the human heart failure cohort.’ It is unclear what is meant by this and where the emerging data is presented. Figure S3A in the sentence thereafter concerns only human heart data.

Response: The ISO-HMDP cohort provided correlations between gene expression and diastolic function (Fig. 5G-5I). The “two-hit” HFpEF cohorts provided genes with sex-biased expression that were induced in response to the HFpEF diet (Fig. S5). The human heart failure cohorts provided genes with sex-biased expression and enriched pathways (Fig. 2D-2G and S6A). These three datasets were overlapped to identify candidate causal genes with sex-biased gene expression.

We have now attempted to clarify this point (page 6): “We also sought to identify sex-biased regulators of diastolic function by overlapping the gene expression data in female ISO HMDP (Fig. 2B-2C) and “two-hit” HFpEF cohorts (Fig. S5), as well as the human heart failure cohort (Fig. 2D-2G) ³. In the human heart failure cohort, sex-biased DEGs were enriched in pathways including cell differentiation and mitochondrion (Fig. S6A). Among the sex-biased genes, *Acsf6* (Acyl-CoA Synthetase Long Chain Family Member 6) expression was significantly lower in female hearts compared with males across the HMDP (Fig. 5B) and was further decreased after HFpEF development (Fig. 5C and 5D).”

We have also substituted “sex-biased” for “sex-specific”.

Page 6: ‘In particular, we focused on a genome-wide association locus for E/A ratio on mouse chromosome 8.’ I suppose you mean genome-wide significant?

Response: We meant significant genome-wide association locus. The locus was significant when corrected for multiple comparisons and was identified by genetic association. We have now added a locus zoom plot of the region associated with E/A ratio (Fig. S7).

Figure 5: The arrow in panel F points to chr11 whereas in the text on page 6 the authors talk of chr8 (line 250). The mouse *Acs16* gene is indeed on chr11. Please correct text.

Response: Thank you. We have corrected this in the main text, but we want to emphasize that *Acs16* was identified based on the correlation of the *cis*-eQTL with diastolic function rather than E/A locus overlap.

Page 6: ‘Gonadectomized mice exhibited reduced *Acs16* expression in the heart after HFpEF (Fig. 5D-5E), suggesting its contribution to increased mitochondrial and metabolic dysfunction in HFpEF mice.’ Please clarify in text that these were male mice.

Response: We have clarified this in the main text (page 6): “Gonadectomy reduced *Acs16* expression in male hearts whereas it increased *Acs16* expression in female hearts (Fig. 5E), consistent with a contribution to increased mitochondrial and metabolic dysfunction in HFpEF mice.”

In the last paragraph of the Results one should mention earlier in the text that studies were done in female mice.

Response: We now indicate the sex of the mice in the main text. In addition, we have now also examined the overexpression of *Acs16* in male mice and included the results as Figure 6.

One only discovers in the legend of Figure 5 that the eQTL analysis of *Acs16* was conducted in female mice after 3 weeks of isoproterenol infusion (ISO-HMDP). Such info should be added to the Results. Please check ms throughout for similar instances.

Response: Thank you for pointing this out. We have carefully gone over the manuscript and added the details accordingly: “female ISO-HMDP” and “female mice”.

The authors refer to *Acs16* and other genes as mitochondrial genes whereas they mean genes expressed in mitochondria? While this term may be commonly used it can be confusing in this paper where mitochondrial copy number is also investigated.

Response: We agree that the term “mitochondrial genes” can be confusing in this context. By “mitochondrial genes” we meant to indicate genes that are part of the mitochondrial genome or nuclear genes that encode mitochondrial proteins. We now attempt to make our use of the term clear (page 4-5): “We also examined the expression of mitochondrial genes that are part of the mitochondrial genome or nuclear genes that encode mitochondrial proteins in the heart.” The list of such genes is available from MitoCarta (<https://www.broadinstitute.org/mitocarta/mitocarta30-inventory-mammalian-mitochondrial->

proteins-and-pathways). *Acsl6* is a nuclear gene that encodes a protein (ACSL6) that localizes to the mitochondrial outer membrane. We now refer to ACSL6 as a nuclear encoded mitochondrial protein.

The role of *Acsl6* can be better connected to effects of lower mitochondrial levels in the Discussion.

Response: We have revised our Discussion accordingly (page 9-10): “Fourth, the studies of *Acsl6* provide evidence for a causal connection. The *Acsl6* gene, encoding a mitochondrial protein involved in fatty acid metabolism, was validated by overexpression in the heart using an adeno-associated viral vector. Overexpression improved diastolic function, reduced left ventricular mass, increased running distance, and improved glucose disposal (Fig. 6 and 7). It also improved aspects of mitochondrial function, as judged by oxygen consumption of isolated mitochondria in males but not females (Fig. 6 and 7). One limitation of our study is that *Acsl6* shows some overexpression in other tissues, notably in the liver, the effects of which are unclear. Additional studies will be required to determine its mechanisms of action. The mechanistic link between *Acsl6* and diastolic function may relate to ATP production. The heart requires a tremendous amount of ATP during relaxation as well as contraction, as calcium needs to be sequestered in the endoplasmic reticulum.”

Reviewer #2 (Remarks to the Author):

The manuscript examines the basis for sex differences in HFpEF by harnessing the power of the Hybrid Diversity Mouse Panel. They zeroed in on mitochondrial differences between male and female heart, performed gonadectomies, measured mitochondrial function and identified a protective gene. The general conclusions were that male hearts have more mitochondria and more mitochondrial gene expression than females and that this difference is driven hormonally. These results are important and may be relevant to humans with HFpEF.

I have several comment/suggestions:

1. Since mito DNA is higher in male versus female hearts, if they normalize their mito gene expression data to mito DNA copies, what is seen?

Response: Thank you for the suggestion. We have normalized mitochondrial gene expression to mtDNA content and the results showed that sex-biased gene expression differences remained (Fig. S3). The gene expression values were strain-averaged. We used the 77 strains for which male and female mice were sequenced. DESeq2 analysis was performed using mitochondrial DNA content as one of the terms in the regression. We now discuss the point in the Results (page 5): “After normalization to mtDNA content in each strain, mitochondrial genes and ETC genes remained higher in males (Fig. S3).”

2. The differences between this group's results and those of the Hill lab in their two hit model of HFpEF seem very important as the field has been searching for a HFpEF model

in which females show worse phenotypes than males is really important to "get out there" since the Hill lab did not see the expected female worse female difference. But, If background genetic strain is the explanation, that flies in the face of the high genetic diversity in the human clinical population of HFpEF. So, this aspect needs to be discussed.

Response: We agree that this is an important point, and although the explanation for the differences in HFpEF sex bias is unclear, they could be due to differences in the mouse strains studied. We have discussed this as follows (page 9): “We observed that in the “two-hit” HFpEF model female mice of multiple strains exhibited impaired diastolic function as compared to males. This contrasts with the findings of Tong et al⁴. We suspect that the differences may be explained by the genetic background since they used strain C57BL/6N mice whereas we used strain C57BL/6J mice as well as a number of other strains. Gene-by-sex interactions are difficult to detect in humans but are commonly observed in mice⁵”.

3. The authors talk about mitochondrial function in their results from the beginning, but they are using things other than function as surrogate for function. It isn't until Figure 4 that they actually measure function.

Response: We ordered the findings in that manner simply because that was the order in which we performed the experiments. We have now reorganized the text in response to the reviewers' comments. Rather than discussing mitochondria at the beginning of the Results section we now begin with sex differences in HFpEF.

4. In Figure 5 E, the author should state in the text that they only did gonadectomies in males. Why did they not do them in females as they did throughout the paper.

Response: The reviewer makes a good point and we have now carried out gonadectomy studies in the female mice as well. Consistent with our conclusions, we observed opposite effects on mitochondrial function and diastolic function. We have added this in Fig. 3. We repeated gonadectomy and Seahorse assay in more male mice and combined the data with previous ones (Fig. 3D-3I).

Reviewer #3 (Remarks to the Author):

These authors report that levels of “heart mitochondria” are higher in males than in females in both humans and mice. Mitochondrial DNA copy number and levels of mitochondrial genes are increased by testosterone and decreased by estrogen. The paper goes on to make some inferences regarding myocardial diastolic dysfunction.

These authors have previously reported differences in mitochondrial biomass based on sex in white adipose tissue and liver, so the report here extending this finding to heart is at some level incremental. It is not a “surprising discovery” (first sentence of Discussion).

Response: By “surprising” we simply meant that it is unexpected, given the importance of mitochondria in heart function, that this has not previously been observed. We have now removed the term “surprising discovery”.

The authors often-stated assertion that HFpEF is more common in females is based on old and now outdated literature. Please see Ho et al, Circ Heart Fail. 2013; Brouwers et al, Eur Heart J 2013; Dunlay et al, Nature Reviews 2017; Pandey et al, Circ 2018. The notion of a female predilection to HFpEF has largely been abandoned, or at the very least called into question. Furthermore, even if there were a female predisposition for HFpEF, there is no evidence for a cellular/molecular basis; rather, the fact that females tend to be smaller than males might play a role.

Response: We agree that the question of sex differences in HFpEF is controversial, and we have now attempted to make this point more clearly, citing the literature listed by the reviewer. In the Introduction we now state (page 2-3): “One of the questions we have addressed is the role of sex differences in HFpEF. Consistent epidemiological data demonstrate that women are about twice as likely to develop HFpEF as compared to men and that they tend to present with more symptom burden, including significantly increased diastolic dysfunction and enhanced left ventricle stiffness^{6,7,8,9}. However, HFpEF is highly heterogeneous and the notion of sex differences in HFpEF has been called into question^{10,11,12,13}. Compared to men, women have structural and functional differences of the heart including smaller left ventricle (LV) chambers, lower stroke volumes, more prominent concentric remodeling upon pressure overload, higher systolic and diastolic LV stiffness at a given age, and a more profound response to hypertension and obesity^{14,15}. One approach to help resolve the issue is to examine mouse models where genetic and environmental factors can be controlled.”

Our findings support a molecular basis of sex differences in the “two-hit” mouse model of HFpEF, with females being more susceptible. Also, the sex differences in mitochondrial gene expression in human hearts raise the possibility that this may also be true in humans. Thus, our findings support a molecular basis for a sex difference but, as we now make clear, this is not definitive in humans. Our results in no way argue that other functions, such as the size of the heart, are not important. Beyond the sex differences, our results indicate an important role of mitochondrial genes and functions in diastolic function.

I do not understand the leap from mitochondrial biomass to diastolic dysfunction. How do the authors suggest that they are linked mechanistically? Furthermore, HFpEF is marked by a phenotype with numerous features well beyond just diastolic dysfunction. Indeed, throughout the paper the authors seem to equate diastolic dysfunction with HFpEF which is unequivocally wrong.

Response: Regarding the mechanistic link between mitochondria and diastolic function, the heart needs a tremendous amount of ATP during relaxation as well as contraction, as calcium needs to be sequestered in the endoplasmic reticulum. Other factors, such as metabolic control and reactive oxygen species, could also be involved. The link between mitochondrial function and

heart function has previously been studied by a number of groups^{16, 17, 18, 19}. These points are now included in the extensively revised Discussion (page 9-10).

We did not intend to equate diastolic function with HFpEF, and our report includes traits such as lung mass, heart weight and running distance. We have revised our manuscript to make clearer that HFpEF is a heterogeneous disorder and that our studies deal with simplified phenotypes in an animal model.

The authors argue that their studies of “heart mitochondria” reflect cardiomyocyte mitochondria given that mitochondrial biomass in these cells is quite high. That said, the observation remains indirect. The findings should be corroborated in isolated cardiac myocytes.

Response: Thank you for the suggestion. We have now isolated adult cardiomyocytes from both male and female mice after the induction of HFpEF. The mitochondrial respiration is higher in males than females (Fig. 4H-4I), consistent with the results from isolated mitochondria.

The data to correlate mitochondrial biomass with cardiac phenotypes are undoubtedly driven by outlier data points, and hence they are unconvincing (Figs 2D-2H).

Response: Reviewer 1, point 3, was also concerned that the data are driven by outlier points. The correlations were performed using biweight midcorrelation (bicor), which is a robust alternative to Pearson correlation that is not as affected by outliers because the underlying calculations are based on medians and not means. None of the results were considered outliers when using median absolute deviations (a robust alternative to Z-scores). In addition, none of the Z-scores exceeded the maximum theoretical Z-scores for this sample size (<https://cran.r-project.org/web/packages/outliers/index.html>). As suggested by reviewer 1, we have now reanalyzed the data following removal of samples at the extreme high mtDNA content. In most cases the correlations remained statistically significant or suggestive (Fig. S2). Notably, we have corrected the original correlations by filtering out the control mice from the isoproterenol treated mice. The updated plots are shown in Fig. 1I-1L and S2.

To assert that *Acsf6* levels are “dramatically lower” in females based on data in Figure 5B is a huge over-statement. And to infer from these data that *Acsf6* may be a “causal driver” is yet another stretch.

Response: We agree that the word “dramatic” is not appropriate and have removed it. By “causal driver” we simply meant that *Acsf6* was predicted to be causal for diastolic dysfunction based on the fact that the local (or “cis”) regulation of the gene expression correlated with the phenotype. We have removed the word “driver”, which was not meant to signify anything other than a causal relationship. The method is outlined in Gusev et al, *Nature Genetics*, 2016¹ and Seldin et al, *Cell Metabolism*, 2019². Of course, such modeling only suggests candidates and experimental validation is required.

The manuscript is laid out in an atypical manner, including presentation of the study

findings in the Introduction rather than using that section of the paper to lay out the knowns and unknowns and the questions to be addressed in the study. Also, the text is quite redundant in several places. The Discussion is largely a recapitulation of the Results.

Response: We thank the reviewer for pointing this out. We have extensively revised the Introduction, Results, and Discussion (please see Track Changes). The Introduction now states the major questions and the Discussion focuses on conclusions rather than restating the results.

On page 5, the authors cite a paper reporting that female mice are less susceptible to HFpEF than males, but the paper reporting this is not cited.

Response: We apologize for the error and we have corrected this in the text (page 3 and 9): Tong et al⁴.

The term “female mitochondria” is odd and should be avoided.

Response: We agree. We have changed this to “mitochondria in females”.

References

1. Gusev A, *et al.* Integrative approaches for large-scale transcriptome-wide association studies. *Nat Genet* **48**, 245-252 (2016).
2. Seldin M, Yang X, Lusis AJ. Systems genetics applications in metabolism research. *Nat Metab* **1**, 1038-1050 (2019).
3. Cordero P, *et al.* Pathologic gene network rewiring implicates PPP1R3A as a central regulator in pressure overload heart failure. *Nat Commun* **10**, 2760 (2019).
4. Tong D, *et al.* Female Sex Is Protective in a Preclinical Model of Heart Failure With Preserved Ejection Fraction. *Circulation* **140**, 1769-1771 (2019).
5. Norheim F, *et al.* Gene-by-Sex Interactions in Mitochondrial Functions and Cardio-Metabolic Traits. *Cell Metab* **29**, 932-949 e934 (2019).
6. Duca F, *et al.* Gender-related differences in heart failure with preserved ejection fraction. *Sci Rep* **8**, 1080 (2018).
7. Beale AL, Meyer P, Marwick TH, Lam CSP, Kaye DM. Sex Differences in Cardiovascular Pathophysiology: Why Women Are Overrepresented in Heart Failure With Preserved Ejection Fraction. *Circulation* **138**, 198-205 (2018).

8. Kessler EL, Rivaud MR, Vos MA, van Veen TAB. Sex-specific influence on cardiac structural remodeling and therapy in cardiovascular disease. *Biol Sex Differ* **10**, 7 (2019).
9. Ferreira RG, Worthington A, Huang CC, Aranki SF, Muehlschlegel JD. Sex differences in the prevalence of diastolic dysfunction in cardiac surgical patients. *J Card Surg* **30**, 238-245 (2015).
10. Brouwers FP, *et al.* Incidence and epidemiology of new onset heart failure with preserved vs. reduced ejection fraction in a community-based cohort: 11-year follow-up of PREVENT. *Eur Heart J* **34**, 1424-1431 (2013).
11. Pandey A, *et al.* Sex and Race Differences in Lifetime Risk of Heart Failure With Preserved Ejection Fraction and Heart Failure With Reduced Ejection Fraction. *Circulation* **137**, 1814-1823 (2018).
12. Dunlay SM, Roger VL, Redfield MM. Epidemiology of heart failure with preserved ejection fraction. *Nat Rev Cardiol* **14**, 591-602 (2017).
13. Ho JE, *et al.* Predictors of new-onset heart failure: differences in preserved versus reduced ejection fraction. *Circ Heart Fail* **6**, 279-286 (2013).
14. Kuch B, *et al.* Gender specific differences in left ventricular adaptation to obesity and hypertension. *J Hum Hypertens* **12**, 685-691 (1998).
15. Redfield MM, Jacobsen SJ, Borlaug BA, Rodeheffer RJ, Kass DA. Age- and gender-related ventricular-vascular stiffening: a community-based study. *Circulation* **112**, 2254-2262 (2005).
16. Kumar AA, Kelly DP, Chirinos JA. Mitochondrial Dysfunction in Heart Failure With Preserved Ejection Fraction. *Circulation* **139**, 1435-1450 (2019).
17. Brown DA, *et al.* Expert consensus document: Mitochondrial function as a therapeutic target in heart failure. *Nat Rev Cardiol* **14**, 238-250 (2017).
18. Bayeva M, Gheorghiade M, Ardehali H. Mitochondria as a therapeutic target in heart failure. *J Am Coll Cardiol* **61**, 599-610 (2013).
19. Yue P, *et al.* Association between mitochondrial DNA copy number and cardiovascular disease: Current evidence based on a systematic review and meta-analysis. *PLoS One* **13**, e0206003 (2018).

REVIEWERS' COMMENTS

Reviewer #1 (Remarks to the Author):

The authors have clarified all points concerning analysis. Furthermore, the manuscript has been revised for increased clarity and detail.

The reviewer however still has a remaining comment concerning line 168 (In most cases the correlations remained significant or suggestive (Fig. S2)). The reviewer finds that one cannot talk of 'most' here. The sentence is vague. A more transparent presentation of the findings is needed.

Response: Thank you for the comment. We have changed the text accordingly (line 163, page 4):

“With the exception of mitral inflow E velocity, the correlations remained significant (Fig. S2)”

Reviewer #2 (Remarks to the Author):

The revisions to the manuscript have been responsive and I have no further concerns.

Response: We thank the reviewer for helpful criticisms and comments.